# FAP106 is an interaction hub for assembling microtubule inner proteins at the cilium inner junction

Michelle M. Shimogawa [1], Angeline S. Wijono[1], Hui Wang [1,2,3], Jiayan Zhang[1,3,4], Jihui Sha[5], Natasha Szombathy[1], Sabeeca Vadakkan[1], Paula Pelayo[1], Keya Jonnalagadda[1], James Wohlschlegel[5], Z. Hong Zhou [1,3,4] ✉ & Kent L. Hill [1,3,4] ✉

Motility of pathogenic protozoa depends on flagella (synonymous with cilia) with axonemes containing nine doublet microtubules (DMTs) and two singlet microtubules. Microtubule inner proteins (MIPs) within DMTs influence axoneme stability and motility and provide lineage-specific adaptations, but individual MIP functions and assembly mechanisms are mostly unknown. Here, we show in the sleeping sickness parasite *Trypanosoma brucei*, that FAP106, a conserved MIP at the DMT inner junction, is required for trypanosome motility and functions as a critical interaction hub, directing assembly of several conserved and lineage-specific MIPs. We use comparative cryogenic electron tomography (cryoET) and quantitative proteomics to identify MIP candidates. Using RNAi knockdown together with fitting of AlphaFold models into cryoET maps, we demonstrate that one of these candidates, MC8, is a trypanosome-specific MIP required for parasite motility. Our work advances understanding of MIP assembly mechanisms and identifies lineage-specific motility proteins that are attractive targets to consider for therapeutic intervention.

Cilia (synonymous with eukaryotic flagella) are conserved, microtubule-based structures that provide motility and signaling functions at the surface of eukaryotic cells. Cilium motility is required for normal human reproduction, development, physiology, and to combat infection[1]. Motile cilia also drive cell propulsion in eukaryotic pathogens responsible for tremendous human suffering worldwide[2,3]. Given this broad distribution and central importance to eukaryote cell biology and human health, understanding structural foundations of cilium assembly and motility is of broad interest.

Most motile cilia are built around a "9 + 2" cylindrical axoneme consisting of nine doublet microtubules (DMTs) surrounding a central pair of singlet microtubules. Each DMT is comprised of a complete, 13-protofilament A-microtubule (A-tubule) and a partial, 10-protofilament B-microtubule (B-tubule), with non-tubulin proteins forming an "inner junction" (IJ) filament that connects the A- and B-tubules[4,5]. Recent studies have revealed a previously unappreciated, yet universal feature of motile axonemes, showing that DMTs are not hollow but, instead, are lined with a highly organized network of interconnected "microtubule inner proteins" (MIPs)[6–9]. Structures inside DMTs had been reported several decades ago in flagellated protozoa[10,11], but the diversity and ubiquitous distribution of such structures was not recognized until cryogenic electron tomography (cryoET) enabled high-resolution, 3D structural analysis[8,12]. Studies in organisms from diverse eukaryotic lineages[13–17] indicate MIPs are present in both the

[1]Department of Microbiology, Immunology and Molecular Genetics, University of California Los Angeles, Los Angeles, CA 90095, USA. [2]Department of Bioengineering, University of California Los Angeles, Los Angeles, CA 90095, USA. [3]California NanoSystems Institute, University of California Los Angeles, Los Angeles, CA 90095, USA. [4]Molecular Biology Institute, University of California Los Angeles, Los Angeles, CA 90095, USA. [5]Department of Biological Chemistry, University of California Los Angeles, Los Angeles, CA 90095, USA. ✉e-mail: Hong.Zhou@UCLA.edu; kenthill@microbio.ucla.edu

A- and B-tubules and occur in a regularly repeating pattern along the length of the DMT in all motile axonemes. Their broad taxonomic distribution indicates MIPs are fundamentally important for axoneme function and likely were present prior to eukaryote diversification. Meanwhile, absence of regularly spaced MIP structures in non-motile cilia[18,19] suggests they are required specifically for motile axonemes. Recent work has shown MIPs can influence axoneme stability, structure and beating[13,15,20–24]. Nonetheless, many aspects of MIP functions and assembly mechanisms remain poorly understood.

In recent, groundbreaking work, cryogenic electron microscopy (cryoEM) was used to reconstruct near-atomic resolution structures of the axoneme, thereby allowing in situ identification and atomic modeling of MIPs in *Chlamydomonas*, *Tetrahymena*, and mammals[13,15–17,21]. These studies have expanded opportunities for studying MIP function and revealed that while some MIPs are conserved across diverse taxa, others represent lineage-specific adaptations[13,15–17,21]. Axonemes of diverse organisms share general architecture and beating mechanism, and lineage-specific MIPs therefore offer a potential mechanism for meeting organism-specific motility needs[2,14,25], such as those presented by movement of parasitic protozoa through tissues of their human and animal hosts[26,27]. Identification and analysis of lineage-specific MIPs are therefore of great interest.

African trypanosomes, e.g., *Trypanosoma brucei*, and related kinetoplastids *Trypanosoma cruzi* and *Leishmania* spp., are flagellated, protozoan pathogens that cause extensive human suffering and limit economic development in some of the world's poorest regions[28,29]. Trypanosomes also represent diverse eukaryotic lineages that have proven useful for uncovering novel features of eukaryote biology[30,31]. *T. brucei* motility exhibits several distinctive features that support movement through host tissues[14,32] and is required for pathogenesis[33] and transmission[34]. The *T. brucei* flagellum has a large and diverse repertoire of MIP structures, including several that are lineage-specific, particularly in the B-tubule[14]. Except for a few conserved examples, however, proteins comprising most *T. brucei* MIP structures are unknown, representing a critical knowledge gap in *T. brucei* flagellum assembly and motility.

Here, we investigate MIP function and assembly in *T. brucei* and show that FAP106 is required for parasite motility and normal flagellum length. Combining cryoET and tandem mass tag (TMT) quantitative proteomics, we further show that *FAP106* knockdown results in loss of conserved IJ MIP structures and their corresponding proteins, as well as several MIP structures and proteins that are lineage-specific. One of these, MC8, is demonstrated by cryoET to be a lineage-specific *T. brucei* MIP and is required for parasite motility. Together, our results demonstrate that FAP106 is a critical interaction hub required for assembly of conserved and lineage-specific MIPs at the inner junction. This work advances understanding of MIP assembly mechanisms and provides insight into parasite-specific MIPs that are important for *T. brucei* motility. Because they are not present in the mammalian host, such MIPs are attractive for consideration as targets for therapeutic intervention in trypanosome infections.

## Results

### FAP106 is required for parasite motility

FAP106/Enkur is a conserved B-tubule MIP that forms a "tether loop" structure spanning the inner junction (IJ) and interconnecting several other MIPs[13,15,21,35]. Enkur is required for motility of mammalian sperm[36], but the structural foundation of the motility defect is unknown, and the role of FAP106/Enkur in assembly of the IJ and other MIPs has not been examined. We used tetracycline-inducible RNAi[37] to deplete *FAP106* in procyclic form cells, hereafter referred to as "*FAP106* KD". RNAi knockdown reduced *FAP106* mRNA expression by more than 95% (Fig. 1A) without affecting parasite growth rate (Fig. 1B). *FAP106* KD parasites did not exhibit any gross morphological defects, although their flagella were shorter than normal (Fig. 1C–E). Notably, despite

having a beating flagellum, *FAP106* KD parasites rarely exhibited directional motility, even at high cell densities, which typically result in faster directional motility (Fig. 1F, G and Supplementary Fig. 1). Thus, FAP106 is required, either directly or indirectly, for normal parasite motility.

### FAP106 is required for assembly of conserved and lineage-specific MIP structures

To determine whether FAP106 is required for assembly of other MIP structures, we performed cryoET on purified, demembranated flagella from control and *FAP106* KD parasites, processed in parallel. Most known MIPs assemble with a periodicity of 16, or 48 nm[9,13–15,17,20,21,35,38] and we therefore performed sub-tomogram averaging to determine the 48-nm repeating unit of the DMT (Fig. 2, Supplementary Fig. 2 and Supplementary Movie 1) for comparison to the published structures of the 48-nm repeat from *T. brucei*[14], as well as other organisms[13,15,21]. The DMT structure in control samples was in good agreement with published structures[14], although we did not clearly resolve the ponticulus, a MIP structure that bisects the B-tubule lumen of mature, but not nascent flagella[39], likely due to averaging from the mixture of old and new flagella in the sample.

MIP densities in the A-tubule appeared largely unaffected in the *FAP106* KD relative to the control and, given the position of FAP106 in the B-tubule, we focused our analysis on the B-tubule. Cross-sectional views of the cryoET tomogram show that several major B-tubule MIP densities are missing in the *FAP106* KD (Fig. 2B). Although we do not have sufficient resolution to identify the corresponding proteins directly, we were able to infer the identity of conserved MIPs by referencing their position in other organisms (Supplementary Fig. 3)[13,15,17,21]. Longitudinal and cross-sectional views show loss of densities corresponding to the "tether loop" structure of FAP106 in algae and mammals[13,15,21] (Fig. 2B–D, red, Supplementary Movie 1 and Supplementary Fig. 3), consistent with the idea that FAP106 structure is conserved across species[13]. A density corresponding to the expected position of FAP52[15,21,22] (Supplementary Fig. 3) appears largely unaffected in the *FAP106* KD (Fig. 2B–D, F, tan and Supplementary Movie 1). We did not resolve filamentous densities corresponding to FAP45 on protofilaments B6-B9 in other organisms[13,15] (Supplementary Fig. 3), but these protofilaments were heavily decorated in *T. brucei* control cells and relatively devoid of MIP densities in *FAP106* KD (Fig. 2D, F and Supplementary Movie 1). Interestingly, the *FAP106* KD also shows loss of FAP106-associated densities that resemble FAP210 in other organisms (Fig. 2B, D, F, green, Supplementary Movie 1 and Supplementary Fig. 3)[13,15,21,35], although a clear FAP210 homolog could not be identified in *T. brucei* based on sequence or structure-based homology searches[40] (Table 1). Overall, our results indicate that FAP106 is critical for assembly of a subset of B-tubule MIP structures in the vicinity of the IJ.

Beyond conserved MIP structures defined in other organisms, we observed additional densities that were reduced in the *FAP106* KD, which correspond to MIP structures that are not present in all organisms. These lineage-specific MIP structures include a protruding density on protofilament B8, "MIP B8", protruding densities attached to protofilament A12, "MIP A12", and a protruding density on protofilament B5, "MIP B5" (Fig. 2B–D, F, Supplementary Movie 1 and Supplementary Fig. 3). Thus, FAP106 is required for assembly of universally-conserved, as well as lineage-specific MIPs.

### FAP106 is required for IJ filament assembly

The IJ filament is a conserved non-tubulin structure that connects protofilaments A1 and B10 in organisms with cilia[8,19]. The IJ filament is comprised of PACRG and FAP20 subunits that alternate each 4 nm along the axoneme[4,5]. Interestingly, most organisms have a single hole in the IJ filament every 96 nm, corresponding to one missing PACRG subunit per 96-nm repeating unit[8,12,38]. *T. brucei* is unusual in having two holes per 96-nm repeating unit[14] (Fig. 2E and Supplementary Fig. 4). The function of the IJ hole is not known, nor is the structural

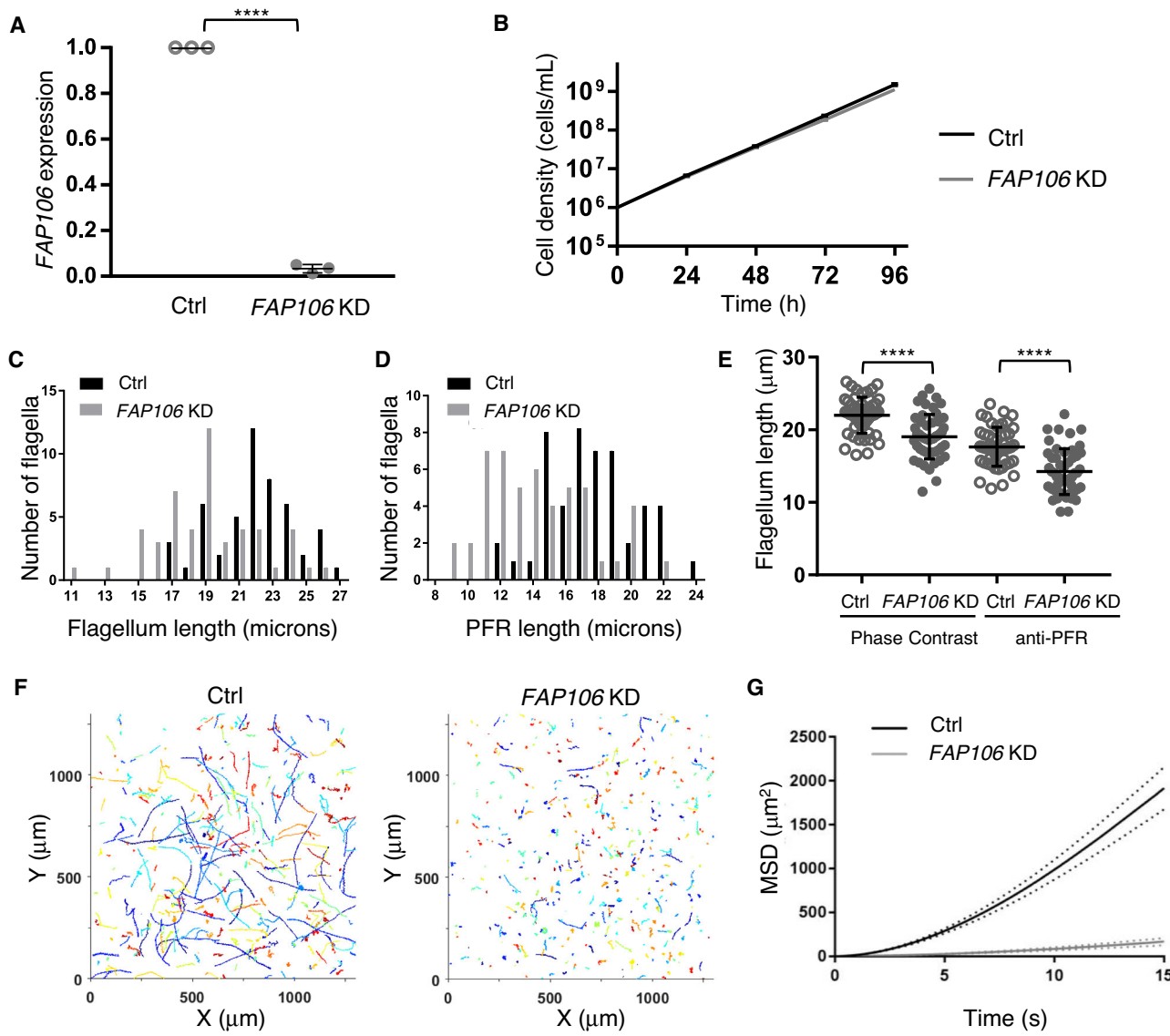

**Fig. 1 | FAP106 is required for parasite motility.** 29-13 (Ctrl) and *FAP106* knockdown (*FAP106* KD) parasites were grown in the presence of tetracycline to induce knockdown. **A** qRT-PCR analysis of *FAP106* mRNA levels. Graph shows the mean ± standard deviation from three independent biological replicates. Unpaired, two-tailed *t*-test **** *p* value ≤ 0.0001. **B** Cumulative growth curve shows the mean cell density ± standard deviation vs. time from three independent biological replicates. **C** Histogram shows the length distribution of purified, demembranated flagella. **D** Histogram shows the distribution of paraflagellar rod (PFR) lengths as measured by anti-PFR immunofluorescence microscopy in detergent-extracted cytoskeletons. **E** Graph shows the mean ± standard deviation of flagellum lengths in (**C**) (Phase Contrast) and (**D**) (anti-PFR). Phase contrast: Ctrl = 22.0 ± 2.5 μm, *FAP106* KD = 19.0 ± 3.1 μm; anti-PFR: Ctrl = 17.6 ± 2.7 μm, *FAP106* KD = 14.2 ± 3.2 μm. *N* = 50 flagella for each. Unpaired, two-tailed *t*-test *****p* < 0.0001. **F** Motility tracks of individual parasites. **G** Mean squared displacement (MSD) of parasites tracked in (**F**). Dotted lines indicate the upper and lower bounds of the standard error of the mean. Ctrl *N* = 489; *FAP106* KD *N* = 675. Data from an independent biological experiment are shown in Supplementary Fig. 1. Source data are provided as a Source Data file.

basis for the presence of two holes per 96 nm in *T. brucei*. Strikingly, our cryoET data show that loss of FAP106 results in two additional holes per 48-nm repeat (Fig. 2E), giving six holes per 96-nm repeat (Fig. 2E, Supplementary Fig. 4 and Supplementary Movie 1). These holes correspond to loss of every fourth subunit of the IJ filament (light blue in Fig. 2E), which likely correspond to PACRG, based on analogy with the *Chlamydomonas* structure[15].

**TMT proteomics identifies FAP106-dependent, lineage-specific MIP proteins**

To identify proteins corresponding to FAP106-dependent MIP densities, we used tandem mass tag (TMT) quantitative proteomics to compare the protein composition of purified, demembranated flagella from control and *FAP106* KD parasites. We quantified more than 2500

proteins and only eight of these were significantly reduced in flagella from *FAP106* KD parasites (≥2-fold reduced, adjusted *p* value ≤ 0.06) (Fig. 3A, Table 2 and Supplementary Data 1). Three proteins were observed to be increased in the knockdown (Fig. 3A and Supplementary Data 1) and were not studied further.

As expected, the protein most reduced in *FAP106* KD parasites was FAP106 itself (Fig. 3A and Table 2), confirming efficient knockdown reported by qRT-PCR (Fig. 1A) and loss of FAP106 protein from the axoneme (Fig. 2). Two other characterized proteins, FAP45 and PACRG-B, were also reduced in the *FAP106* KD (Fig. 3A, B and Table 2). FAP45 was almost 4-fold reduced in *FAP106* KD flagella (Fig. 3A and Table 2), supporting results from cryoET that show reduction of densities near the expected location of FAP45 (Fig. 2, Supplementary Fig. 3 and Supplementary Movie 1) and indicating FAP106 is required for assembly of

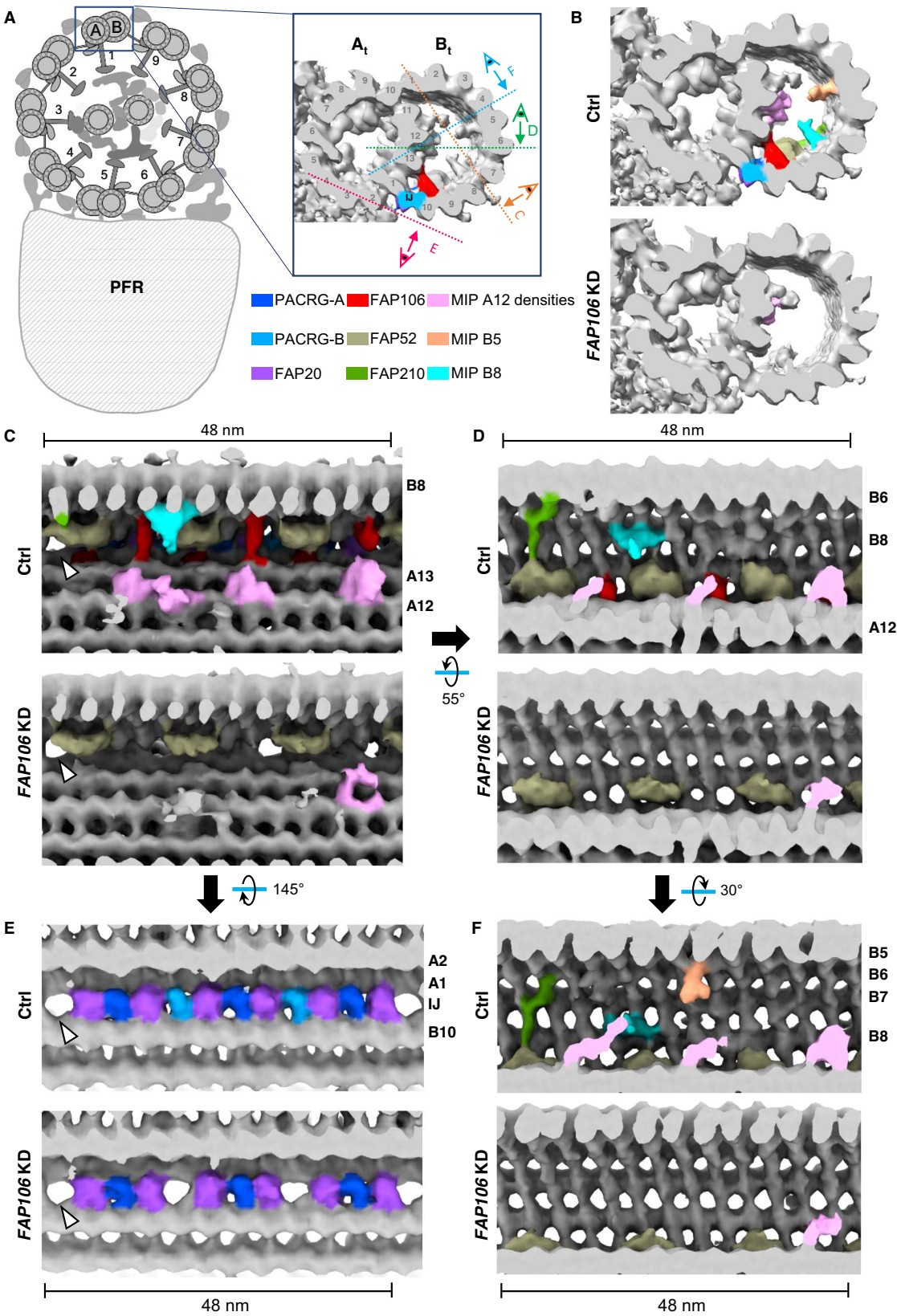

FAP45 into the axoneme. FAP52 and FAP20 were unaffected (Supplementary Data 1), further supporting conclusions from cryoET that these proteins assemble independently of FAP106 (Fig. 2). Notably, *T. brucei* has two PACRG homologs, PACRG-A and PACRG-B (Table 1), which have been suggested to be functionally redundant[41,42]. Despite a high degree of similarity, however, PACRG-A and PACRG-B are readily

distinguishable by mass spectrometry and our TMT proteomics analysis shows that only PACRG-B is lost in the *FAP106* KD (Fig. 3A, Table 2 and Supplementary Data 1). Loss of PACRG-B and not PACRG-A indicates that the extra holes in the *FAP106* KD IJ filament (Fig. 2E and Supplementary Fig. 4) correspond to missing PACRG-B subunits and demonstrates that PACRG-A and PACRG-B are not redundant. Based on the

**Fig. 2 | FAP106 is required for assembly of conserved and lineage-specific MIP structures.** Demembranated flagella were purified from 29-13 (Ctrl) and *FAP106* knockdown (*FAP106* KD) parasites grown in tetracycline to induce knockdown. Structure of the 48-nm repeat of the DMT was obtained by cryoET, with sub-tomographic averaging. Densities are shown as surfaces and colored according to the legend. Conserved MIPs are colored based on similarity to published structures[13,15,21] (Supplementary Fig. 3). Lineage-specific densities (MIP A12, MIP B5, MIP B8) not present in *Chlamydomonas* or bovine structures are colored to highlight densities that are substantially reduced in *FAP106* KD. **A** Schematic illustrating a cross-section of the "9 + 2" axoneme and paraflagellar rod (PFR) viewed from flagellum tip to base. Adapted from[75]. Enlarged panel shows a single DMT, with A- and B-tubules indicated ($A_t$, $B_t$) and protofilaments numbered. FAP106 (red) and the inner junction filament (IJ) comprised of alternating PACRG (blue) and FAP20 (purple) are colored for reference. Viewing angles shown in (**C–E**) are indicated.

**B** Cross-sectional view of the DMT. Colored densities indicate structures reduced or missing in *FAP106* KD compared to Ctrl. **C, D** Longitudinal views of the DMT. Viewing angles are indicated in (**A**), inset. The 48-nm repeat is indicated. Note that *T. brucei* has a structure similar to FAP210 (green) despite not having a clear FAP210 homolog (Table 1). White arrowheads indicate the position of the single hole found within each 48-nm repeat in Ctrl DMTs. **E** Longitudinal views of the DMT, showing the IJ filament. Viewing angles are indicated in (**A**), inset. The 48-nm repeat is indicated. White arrowheads indicate the position of the single hole found within each 48-nm repeat in Ctrl DMTs. FAP20 and PACRG subunits are colored according to the legend. PACRG-A and PACRG-B are colored to be consistent with TMT proteomics data showing that only PACRG-B is lost in *FAP106* KD (Fig. 3A and Supplementary Data 1), which suggests the extra holes in the *FAP106* KD structure correspond to the position of PACRG-B in control axonemes. **F** Longitudinal view showing reduction of MIP B5 in *FAP106* KD.

position of the extra holes in the *FAP106* KD, specific loss of PACRG-B, and a recent study demonstrating alternating PACRG-A/B homologs in *Tetrahymena*[17], the two IJ holes per 96-nm repeat in control parasites most likely also correspond to missing PACRG-B.

In addition to the known MIPs described above, five previously uncharacterized proteins were reduced in flagella from *FAP106* KD parasites (Fig. 3A and Table 2). One of these proteins (RPC19, Tb927.11.8890) localizes to the nucleus[43,44], suggesting it is unlikely to be a MIP. The remaining four proteins (Tb927.10.7120, Tb927.11.4920, Tb927.11.2770, and Tb927.3.3200), are kinetoplastid-specific and localize to the flagellum (Supplementary Fig. 5)[43,44], making them strong candidates for lineage-specific, FAP106-dependent MIPs. Importantly, these four proteins were separately identified as "MIP candidates", MC3, MC5, MC8, and MC15, in a completely independent APEX2-based proximity proteomics analysis as being in proximity to B-tubule MIPs FAP45 and FAP52 (detailed in Supplementary Methods; Supplementary Fig. 6 and Supplementary Data 2). APEX2 proximity proteomics identifies proteins in proximity to an APEX2-tagged protein, "the bait", based

on biotinylation and subsequent streptavidin purification and proteomic identification[45,46]. Using B-tubule MIPs FAP45 and FAP52 as APEX2-tagged bait, we identified 15 kinetoplastid-specific proteins, which were named MC1-15. Identification of MC3, MC5, MC8, and MC15 as FAP106-dependent and adjacent to known B-tubule MIPs greatly supports their assignment as kinetoplastid-specific MIPs.

To test the requirement of MC3, MC5, MC8 and MC15 for flagellum structure and function, we used tetracycline-inducible RNAi to individually knockdown each of these proteins. Knockdown, confirmed by loss of the respective proteins from the flagellum (Supplementary Fig. 5), did not have substantial effects on parasite growth. Loss of MC5 did not affect parasite motility, while MC3, MC8, and MC15 are each required for normal motility (Fig. 3C and Supplementary Fig. 7A). *MC8* KD and *MC15* KD parasites also showed small reductions in flagellum length (Fig. 3D and Supplementary Fig. 7B), while the *MC3* KD did not. To ask whether additional proteins depend on these MCs for assembly into the axoneme, we performed TMT proteomics on flagella isolated from parasites in which the individual *MC*s were knocked down (Supplementary Fig. 8 and Supplementary Data 1). These proteomic analyses show significant reduction of each MC in the respective knockdown, confirming specific knockdown of the target proteins, but no additional proteins were lost (≥2-fold reduced, adjusted *p* value ≤ 0.06) in any of the four knockdowns (Supplementary Fig. 8 and Supplementary Data 1). Therefore, motility defects in *MC3*, *MC8*, and *MC15* knockdowns reflect a requirement for these individual, kinetoplastid-specific proteins in trypanosome motility.

### MC8 is a lineage-specific *T. brucei* MIP that comprises the MIP B8 density

To discover MIPs corresponding to FAP106-dependent MIP densities, we fit the AlphaFold[47–49] models of the above MCs into our cryoET structures. AlphaFold prediction of individual proteins returned low-confidence predictions for MC3, MC5, and MC15, but predicted that MC8 folds into a structure with a high-confidence pyramid-like domain formed by α helices (Fig. 4A), which fits well into the MIP B8 density (Fig. 4B). To determine whether MC8 in fact corresponds to the MIP B8 density, we performed cryoET on purified, demembranated flagella from control and *MC8* KD parasites, processed in parallel (Fig. 4C–E). In contrast to Fig. 2, both samples resolved a more defined ponticulus structure[39] in the B-tubule (Fig. 4C). Importantly, except for this known age-dependent marker[39], there are no major differences between the controls from the two experiments, confirming that structures reduced in the *FAP106* KD (Fig. 2) are due to the knockdown and not age-dependent differences. TMT analysis showed that MC8 is the only protein lost in the *MC8* KD (Supplementary Fig. 8C and Supplementary Data 1), so any density altered in the knockdown must correspond to MC8 itself. The sub-tomographic average of the 48-nm repeating unit from *MC8* KD axonemes was nearly identical to that of controls except for the absence of the MIP B8 density (Fig. 4C–E). Thus, MC8 is a bona fide, kinetoplastid-specific B-tubule MIP that is required for normal

### Table 1 | Known B-tubule MIPs

| *Chlamydomonas* | Mammals | *Tetrahymena* | *Trypanosoma* |
|---|---|---|---|
| [a]PACRG | PACRG | PACRGA; PACRGB; PACRGC | Tb927.3.2310; Tb927.9.9940 |
| [a]FAP20 | CFAP20 | CFAP20 | Tb927.10.2190 |
| FAP45 | CFAP45 | CFAP45 | Tb927.8.4580 |
| FAP52 | CFAP52 | CFAP52A; CFAP52B; CFAP52C | Tb927.11.7560 |
| FAP106 | ENKUR | CFAP106A; CFAP106B; CFAP106C | Tb927.11.4880 |
| FAP77 | CFAP77 | CFAP77A; CFAP77B | |
| FAP210 | CFAP210 | CFAP210 | |
| FAP90 | CFAP90 | | |
| FAP126 | Flattop | | |
| FAP144 | FAM183A | | |
| FAP276 | CFAP276 | | |
| FAP112 | | CFAP112A; CFAP112B | |
| | EFCAB6 | | |
| | SPACA9 | | |
| | | IJ34 | |
| | | OJ2 | |
| | | | Tb927.11.2770 (MIP B8, this study[b]) |

[a]FAP20 and PACRG form the IJ filament.
[b]Corresponding structure was called "MIP3c" in ref. 14 but our current study suggests this structure is distinct from MIP3c (FAP45).

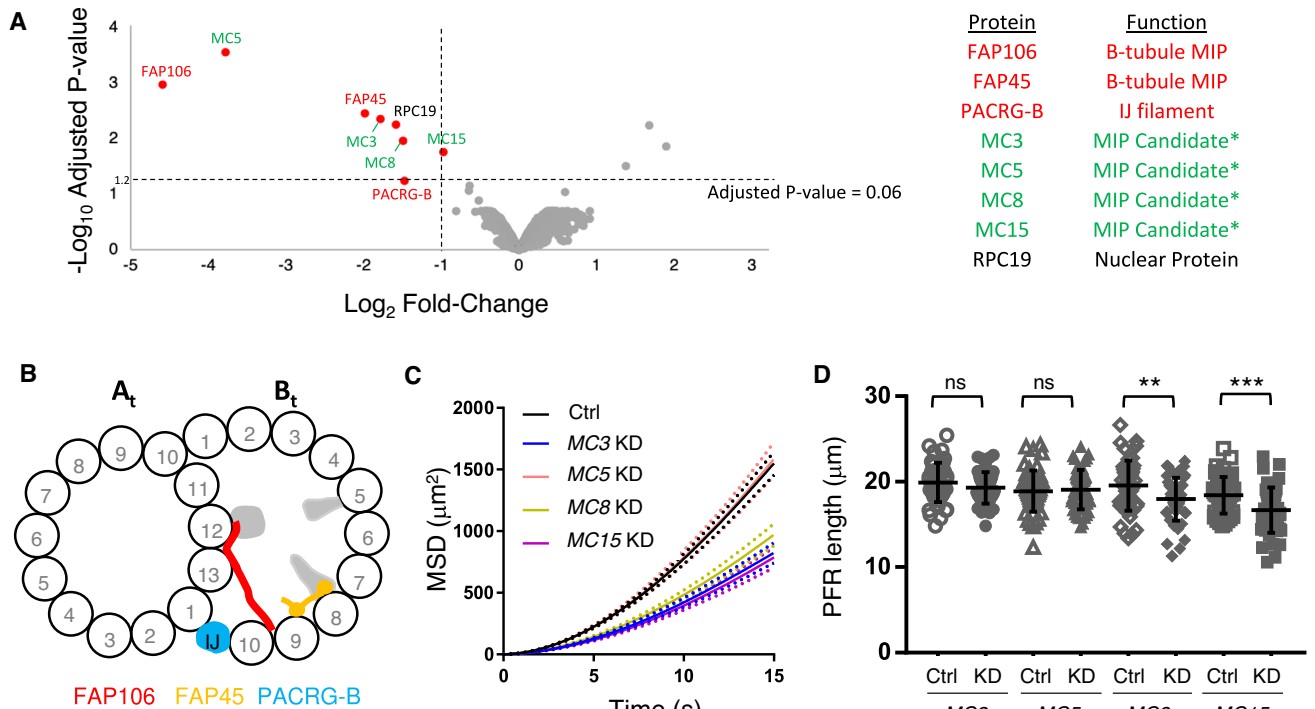

| Protein | Function |
| --- | --- |
| FAP106 | B-tubule MIP |
| FAP45 | B-tubule MIP |
| PACRG-B | IJ filament |
| MC3 | MIP Candidate* |
| MC5 | MIP Candidate* |
| MC8 | MIP Candidate* |
| MC15 | MIP Candidate* |
| RPC19 | Nuclear Protein |

**Fig. 3 | TMT proteomics identifies FAP106-dependent, lineage-specific MIP proteins. A** Demembranated flagella were purified from 29-13 (Ctrl) or *FAP106* KD parasites grown in tetracycline to induce knockdown. Volcano plot shows significance using a moderated two-sided *t*-test with Benjamini–Hochberg adjustment for multiple comparisons (−Log₁₀ Adjusted *p* value) vs. relative abundance in *FAP106* KD/Ctrl (Log₂ Fold-Change) for all proteins quantified by TMT proteomics of two independent biological samples. Proteins meeting the filtering criteria (≥2-fold decrease, adjusted *p* value ≤ 0.06; Table 2) are indicated by red dots; text labels indicate known proteins (red) and putative MIP candidates (green). Source data are provided in the Source Data file and Supplementary Data 1. Asterisks indicate proteins independently identified as putative B-tubule MIP candidates in a separate proximity labeling approach (Supplementary Fig. 6 and Supplementary Data 2). **B** Cross-sectional illustration of the DMT as viewed from the flagellum tip, showing the positions of MIP structures lost in *FAP106* KD, with known MIPs colored and unknown densities in gray, as observed by cryoET (Fig. 2). A- and B-tubules are indicated (A_t, B_t) and protofilaments are numbered. **C** Motility analysis showing mean squared displacement (MSD) of 29-13 (Ctrl) and *MC* KD parasites from ≥2 independent biological replicates. Dotted lines indicate the upper and lower bounds of the standard error of the mean. Ctrl *N* = 1761; *MC3* KD *N* = 627; *MC5* KD *N* = 753; *MC8* KD *N* = 709, *MC15* KD *N* = 751. Individual replicates are shown in Supplementary Fig. 7A. **D** Graph shows the mean ± standard deviation of flagellum lengths as measured by anti-paraflagellar rod (PFR) immunofluorescence microscopy on detergent-extracted cytoskeletons prepared from the indicated KD parasites or their respective mNeonGreen (NG)-tagged parental cell lines (Ctrl). *N* = 50 flagella for each, except *N* = 45 flagella for MC15-NG. Unpaired, two-tailed *t*-test: MC3-NG vs. *MC3* KD *p* = 0.14, MC5-NG vs. *MC5* KD *p* = 0.72, MC8-NG vs. *MC8* KD *p* = 0.005 and MC15-NG vs. *MC15* KD *p* = 0.0008. Source data and means are provided as a Source Data file.

parasite motility (Fig. 3C and Supplementary Fig. 7A). The motility defect of *MC8* KD parasites, in which the only missing protein is MC8, demonstrates importance of lineage-specific MIPs to flagellar motility.

## Discussion

### *FAP106* knockdown provides insights into MIP assembly mechanisms

Functional analysis lags behind identification and structural assignment of MIP proteins. Here we combine functional, structural, and proteomic analysis to demonstrate that the conserved MIP FAP106 is required for motility and is a key interaction hub for IJ filament and B-tubule MIP structures. The role of FAP106 as a key interaction hub is emphasized by our finding that, while FAP106 is required for assembly of FAP45 and lineage-specific MIP proteins MC3, 5, 8, and 15 (Figs. 2, 3 and 5), loss of these individual proteins does not affect other MIP proteins or structures (Supplementary Fig. 8, Fig. 4 and Supplementary Data 1[22]). Thus, rather than interdependency, our results suggest a hierarchical mechanism for B-tubule MIP assembly. FAP106-dependency of FAP45 and a FAP210 structural homolog suggests that extensive interactions of FAP45 and FAP210 with the tubulin lattice[13,15] are relatively weak, while interactions with FAP106[13] are critical for stable binding and establishing their periodicity within the 48-nm repeat. This supports the idea that an important role of MIPs is to enable complex assembly of proteins along otherwise uniform

protofilament polymers of tubulin[13,15,17,20] and suggests that loss of FAP45 and FAP210 in *RIB72* mutants[50] may reflect dependency on FAP106 rather than a requirement for RIB72 directly. Our studies also provide a structural explanation for motility defects of *FAP106* mutants in trypanosomes (Fig. 2), and in sperm from *Enkur* mutants in mice[36].

### The IJ filament subunit PACRG exhibits functional heterogeneity in trypanosomes

PACRG alternates with FAP20 to form the highly conserved IJ filament[4,5]. One of the more mysterious features of DMTs is a hole in the IJ filament that occurs once per 96-nm repeat in most organisms and corresponds to a missing PACRG[8,12,38]. *T. brucei* is unusual in having two IJ holes per 96-nm repeat, one at the site of nexin-dynein regulatory complex (NDRC) attachment as observed in other organisms, and one proximal to the NDRC (Supplementary Fig. 4)[14]. The proximal hole is devoid of external structures on the DMT (Supplementary Fig. 4)[14], suggesting it is dictated by MIP interactions rather than NDRC attachment, which is proposed for the NDRC hole[51]. Our findings support this idea, as loss of FAP106 results in extra holes in the IJ filament (Fig. 2E and Supplementary Fig. 4). Some organisms have multiple PACRG homologs[17,41,52]. Two PACRG homologs in *T. brucei*, PACRG-A and PACRG-B, have been suggested to be functionally redundant[41,42]. Our combined structural and proteomics data (Figs. 2E and 3A)

**Table 2 | Proteins significantly reduced in *FAP106* KD flagellum skeletons by TMT proteomics**

| Protein | Gene ID | [a]Fold reduction | [b]Adjusted *p* value | Protein localization | Function |
|---|---|---|---|---|---|
| FAP106 | Tb927.11.4880 | 24.02 | 0.0011 | Flagellum | B-tubule MIP |
| FAP45 | Tb927.8.4580 | 3.95 | 0.0035 | Flagellum | B-tubule MIP |
| PACRG-B | Tb927.9.9940 | 2.77 | 0.0576 | Flagellum | IJ filament |
| MC5 | Tb927.11.4920 | 13.74 | 0.0003 | Flagellum | MIP candidate[c] |
| MC3 | Tb927.10.7120 | 3.44 | 0.0044 | Flagellum | MIP candidate[c] |
| MC8 | Tb927.11.2770 | 2.81 | 0.0111 | Flagellum | MIP candidate[c] |
| MC15 | Tb927.3.3200 | 1.95 | 0.0177 | Flagellum | MIP candidate[c] |
| RPC19 | Tb927.11.8890 | 2.99 | 0.0055 | Nucleus | RNA polymerase |

[a]Source data are provided as a Source Data file.

[b]Moderated two-sided *t*-test with Benjamini–Hochberg adjustment for multiple comparisons.

[c]MIP candidates were identified by independent APEX2-based proximity proteomics using known B-tubule MIPs FAP45 and FAP52 as bait (see Supplementary Data 2).

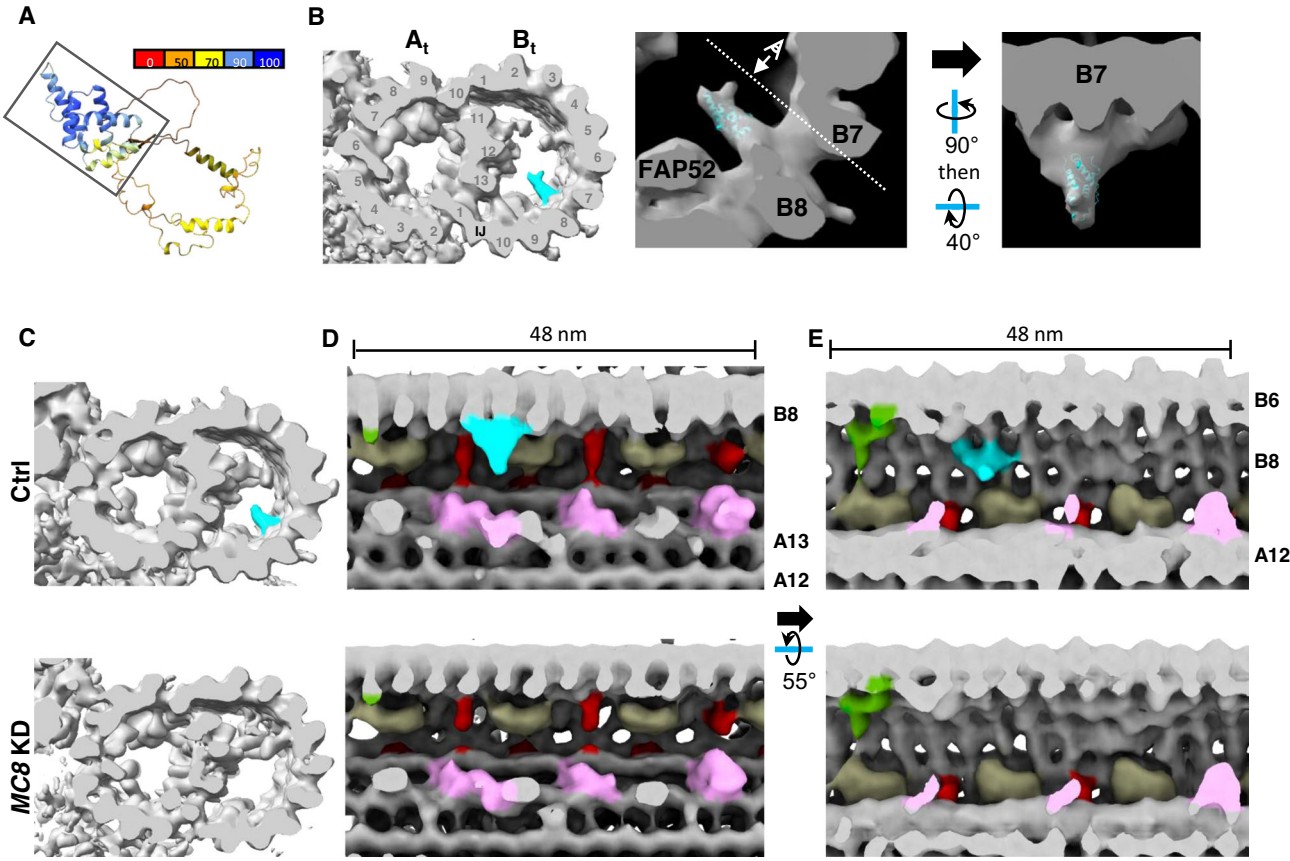

**Fig. 4 | MC8 is a lineage-specific *T. brucei* MIP that comprises the MIP B8 density. A** AlphaFold model for MC8[47,48], shown as ribbons and colored according to confidence. Box indicates high-confidence region modeled in (**B**). **B** MC8 AlphaFold model (ribbon) fitted into MIP B8 cryoET density (gray). (Left) Cross-sectional view of the DMT cryoET density map, with the MIP B8 density colored aqua. (Middle, Right) Enlarged view of MIP B8 density with AlphaFold model shown in aqua. Viewing angle shown in Right panel is indicated by the dashed line in Middle panel. **C–E** Demembranated flagella were purified from 29-13 (Ctrl) and *MC8* knockdown (*MC8* KD) parasites grown in tetracycline to induce knockdown. Structure of the 48-nm repeat of the DMT was obtained by cryoET, followed by sub-tomographic averaging. **C** Cross-sectional view of the DMT with MIP B8 density colored in aqua. **D**, **E** Longitudinal views of the DMT. Viewing angles and colors are as indicated in Fig. 2C, D.

instead support a model in which *T. brucei* PACRG-A and PACRG-B both regularly assemble within each IJ filament, alternating with FAP20 in turn, and with holes in the IJ filament corresponding to missing PACRG-B, but not PACRG-A. Such an arrangement would be consistent with recent data showing alternating PACRG isoforms in *Tetrahymena*[17]. Distinct locations and assembly requirements for PACRG-A and PACRG-B indicate they play distinct roles in assembly of the IJ. Although PACRG is conserved among ciliated organisms, the family of PACRG proteins shows substantial heterogeneity in the N-terminal region that has been shown to interact closely with the DMT lattice in *Chlamydomonas*[17,21,52]. Divergent N-termini of *T. brucei* PACRG-A vs.

PACRG-B may contribute to the differential dependence on FAP106 and their distinct roles in assembly of the IJ.

## MC8 is a lineage-specific *T. brucei* MIP that is required for parasite motility

The growing body of evidence suggests IJ subunits (PACRG and FAP20) and a core set of conserved MIPs are found in all motile cilia, while other MIPs vary between organisms[13–17,21,35,38]. FAP45, FAP52, and FAP106 represent core B-tubule MIPs while FAP126, FAP276, and FAP210 are only found in some organisms (Table 1). MIP A12 densities (Fig. 2B, C, F) are not present in *Chlamydomonas* or mammals[13,15,21] and

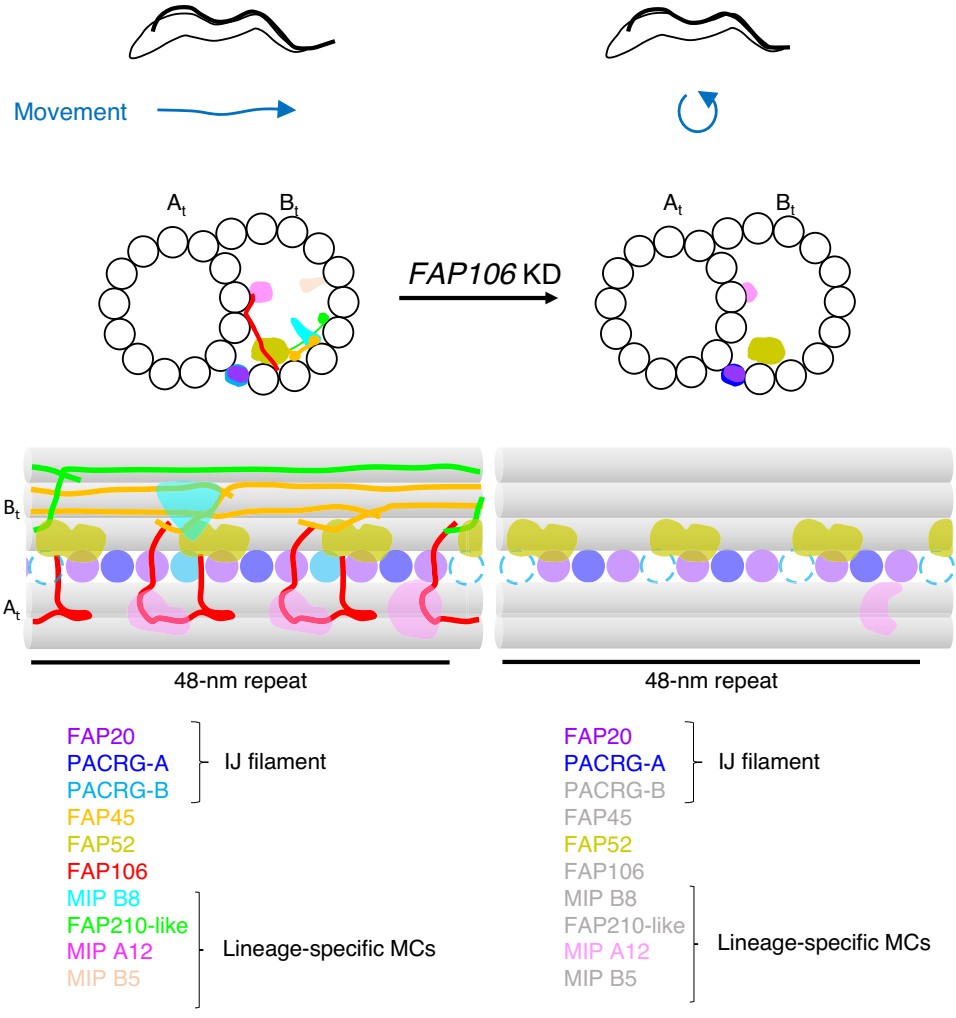

**Fig. 5 | FAP106 is required for parasite motility and assembly of conserved and lineage-specific MIPs at the inner junction.** Normal parasite motility relies on FAP106 as a critical interaction hub for assembly of MIPs at the inner junction (IJ). Cross-sectional and longitudinal views of the DMT are illustrated. Conserved IJ MIPs PACRG-A, FAP20 and FAP52 assemble independently of FAP106, however PACRG-B does not stably assemble into the IJ filament without FAP106. The B-tubule remains intact and connected to the A-tubule at the IJ, despite loss of multiple IJ MIPs. FAP106 is also critical for assembly of the conserved MIP, FAP45, and a FAP210-like protein (modeled as filaments based on published structures[13,15,21]), as well as several previously unknown lineage-specific MIPs.

appear more prominent in *T. brucei* than MIP5a and MIP5b located at a similar position in *Tetrahymena*[9,17,35,38], but the proteins that comprise these densities remain unknown. We demonstrate that MC8 is a bona fide lineage-specific MIP corresponding to the MIP B8 structure (Fig. 4). A density with location and periodicity similar to MIP B8 is described in some studies of *Tetrahymena*[9,17,35,38], but we were unable to identify any protein outside of kinetoplastids with sequence or structural similarity[40,47,49] to the MIP B8 protein, MC8. Loss of MC8 in the *FAP106* KD (Fig. 2B–D) is likely due to loss of FAP45, which runs along protofilaments B7-B8[13,15] and contacts FAP106 directly[13]. Loss of MC8 compromises motility (Fig. 3C and Supplementary Fig. 7A) without affecting other MIPs (Fig. 4 and Supplementary Fig. 8C), demonstrating that lineage-specific MIPs directly impact trypanosome motility. Validation of MC8 as a bona fide *T. brucei* MIP in the current study, through a combination of mutant analysis, structural, functional, and proximity proteomics approaches, demonstrates the power of this combined approach for MIP identification and analysis. Our studies also strongly suggest that MC3, 5, and 15 likewise correspond to lineage-specific *T. brucei* MIPs. Trypanosome motility is necessary for virulence in the mammalian host[33] and transmission through the tsetse fly vector[34]. Therefore, identification of trypanosome-specific MIPs provides attractive targets to consider for treatment of devastating diseases caused by these pathogens.

## Model for assembly of the IJ

Our work provides an important advance toward elucidating the order of MIP assembly and supports the following model for IJ assembly (Fig. 5). FAP20 binds every 8 nm, together with at least a subset of PACRG, to form the IJ filament and this occurs independently from assembly of FAP52[21] and FAP106 (this work). FAP52 binds every 16 nm, independently of the IJ filament[4] and FAP106 (this work), and may connect the IJ filament to the B-tubule via FAP276 or other analogous structure[21,35]. FAP106 also binds every 16 nm, independently from FAP52[21], bridging the A- and B-tubules directly and serving as a critical guidepost for assembling additional MIPs such as FAP45 and FAP210 (this work). Although FAP52 and FAP106 bind regularly within each 48-nm repeat, their connections to other MIPs differ every 16 nm. Despite apparent contacts between FAP52 and other IJ MIPs[13,15,21,35], assembly of IJ MIPs in *Chlamydomonas* is largely unaffected by loss of FAP52[21]. Thus, while FAP52 and FAP106 form contacts with each other and other IJ MIPs, they assemble independently from one another, and FAP106 appears to be more critical for assembly of additional B-tubule MIPs. Going forward, as additional MIPs are identified, particularly those that are lineage-specific, application of combined proteomics, cryoEM, and mutant analysis, as done here, will be necessary to completely define MIP assembly mechanisms and to

yield a full understanding of their contribution to flagellum function.

## Methods

### Biological materials
All unique biological materials are available from the authors upon request.

### *Trypanosoma brucei* culture
Procyclic *T. brucei brucei* (strain 29-13) originally obtained from George Cross (Rockefeller University)[53] were cultivated in SM medium[54] supplemented with 10% heat-inactivated fetal bovine serum (FBS) at 28 °C with 5% CO$_2$.

### In situ tagging
mNeonGreen and APEX2-tagged cell lines were generated in the 29-13 background by C-terminal tagging[55] with pPOTv6-puro-puro-mNG or pPOTv7-blast-blast-APEX2-3xHA, respectively. pPOTv7-blast-blast-APEX2-3xHA contains the same APEX2-3xHA tag as in[46]. For MC3-NG, a transient CRISPR-Cas9 system[56] was used to tag both alleles. See Supplementary Table 2 for list of primers.

### Tetracycline-inducible knockdown
Constructs for tetracycline-inducible knockdown were designed using RNAit[57] and cloned into p2T7-177[37]. See Supplementary Table 2 for list of primers. NotI-linearized plasmids were transfected into 29-13 (*FAP106* KD) or the corresponding MC-NG tagged cell lines (*MC* KDs) using established methods[54]. Clonal lines were generated by limiting dilution. Knockdown was induced by growing cells in the presence of 1 µg/ml tetracycline (Tet) for at least 3 days. Control parasites were grown in the presence of tetracycline for comparison to the respective KDs. For growth curves, cells were counted with a Beckman Coulter Z1 particle counter and diluted daily to a concentration of $1 \times 10^6$ cells/ml, in triplicate.

### qRT-PCR
Quantitative real-time PCR (qRT-PCR) was performed as previously described[58] with the following modifications. *FAP106*-specific primers were designed using RNAit[57]. Analyses were performed in duplicate on three independent RNA preparations and values were normalized to *RPS23*[59]. See Supplementary Table 2 for list of primers.

### Motility analyses
Cells were grown to ~$1 \times 10^6$ cells/ml and loaded into polyglutamate-coated motility chambers[33] sealed with Vaseline. Thirty second videos were acquired under dark-field illumination at ~40 frames per second (fps) using a Hamamatsu ORCA-Flash 4.0 camera on a Zeiss Axio Imager A2 microscope with an EC Plan-Neofluar 10x/0.3 objective lens and Zen 2.6 Pro software. At least five videos were collected for each independent biological replicate. Single cell tracking and mean squared displacement were performed in MATLAB as previously described for fluorescent parasites[33]. For "high density" motility analyses, cells were grown to ~$1 \times 10^7$ cell/ml, then diluted in conditioned medium to ~$1 \times 10^6$ cells/ml and analyzed as above.

### Purification of demembranated flagella (detergent-extracted flagellum skeletons)
Detergent extraction was done to demembranate flagellum skeletons, essentially as described[14]. Briefly, cells were washed in Dulbecco's PBS (DPBS), resuspended in Extraction buffer (20 mM HEPES, pH 7.4, 1 mM MgCl$_2$, 150 mM NaCl, 0.5% IGEPAL CA-630 (NP40), 2x Sigmafast EDTA-free protease inhibitor cocktail) for 15 min at room temperature, followed by depolymerization of subpellicular microtubules with 1 mM CaCl$_2$ on ice for 30 min. Detergent-extracted flagellum skeletons were pelleted at 1500 × g for 10 min at 4 °C.

### Flagellum length measurements
For measurements of purified flagella, detergent-extracted flagellum skeletons were prepared as described above and phase contrast images were acquired on a Zeiss Axioskop II microscope with a Plan-Apochromat 63x/1.4 objective lens using Axiovision 4.8 software. For measurements of the paraflagellar rod (PFR) in detergent-extracted cytoskeletons, cells were washed in DPBS and allowed to adhere to glass cover slips for 10 min. Cover slips were rinsed to remove unattached cells, then extracted with PEME buffer (100 mM PIPES, 2 mM EGTA, 1 mM MgSO$_4$, 0.1 mM EDTA, pH 6.8) + 1% NP40 + 2x Sigmafast EDTA-free protease inhibitor cocktail for 10 min. Cover slips were fixed in methanol at −20 °C for 10 min, allowed to dry, then rehydrated in DPBS for 15 min. Cover slips were blocked in DPBS + 8% normal donkey serum + 2% bovine serum albumin then incubated in rabbit anti-PFR2[60] primary antibody diluted 1:1000 in blocking solution, followed by donkey anti-rabbit Alexa 594 secondary antibody (Invitrogen A21207) diluted 1:1500 in blocking solution. Cover slips were mounted in Vectashield containing DAPI and images were acquired on a Zeiss Axioskop II fluorescence microscope with a Plan-Apochromat 100x/1.4 objective lens and Axiovision 4.8 software or on a Zeiss Axio Imager Z1 fluorescence microscope with a Plan-Apochromat 63x/1.4 objective lens and Zen software. Flagellum lengths were measured in Fiji using the freehand line tool and analyzed in GraphPad Prism 7.

### CryoET
CryoET grids and detergent-extracted flagellum skeletons were prepared essentially as described[14]. Briefly, $2 \times 10^8$ cells from log phase cultures of similar cell density were harvested and washed twice in DPBS. Flagellum skeletons were prepared as described above, resuspended in 100 µl Extraction buffer and spun over a 30% sucrose cushion in Extraction buffer without NP40 at 800 × g for 5 min at 4 °C. In total, 100–150 µl of flagellum skeletons were collected from the upper fraction, washed once with 700 µl Extraction buffer, and once with 1 ml Extraction buffer without NP40. Flagellum skeletons were resuspended in Extraction buffer without NP40 and vitrified on quantifoil grids with 5 nm fiducial gold beads.

With *SerialEM*[61], tilt series were collected in a Titan Krios instrument equipped with a Gatan imaging filter (GIF) and a post-GIF K3 direct electron detector in electron-counting mode with parameters listed in Supplementary Table 1. Frames in each movie of the raw tilt series were aligned, drift-corrected, and averaged with *Motioncor2*[62]. The micrographs in each tilt series were aligned and reconstructed into 3D tomograms by either weighted back projections (WBP, for subtomographic averaging) or simultaneous iterative reconstruction technique (SIRT, for visualization and particle picking) using the *IMOD* software package[63]. The contrast transfer function (CTF) was determined by *ctffind4*[64] and corrected with the *ctfphaseflip* program[65] of *IMOD*.

As detergent-extracted flagellum skeletons from log-phase cultures contain a mixture of old and new flagella that cannot be distinguished due to lack of cell body remnants, the data represent an unbiased, random sampling of axonemes. Data were collected from the middle region of the flagella to minimize variability due to known differences in protein composition between the proximal and distal ends[66]. To ensure observed differences between samples are due to knockdown of the target protein, rather than age-dependent differences, such as the ponticulus[39], comparisons were done only on samples prepared at the same time from cultures maintained at equivalent cell densities and processed identically, in parallel. Validation of this experimental design is provided by sub-tomogram averaging of particles from individual flagella as described in "Sub-tomogram averaging" below.

### Sub-tomogram averaging, visualization and modeling
4x binned SIRT reconstructed tomograms were further processed by IsoNet[67] to improve the contrast for particle picking. Because of their distinct features, radial spokes were picked as particles to represent

the centers of 96-nm axonemal repeat units. These particles of 96-nm repeat units were subject to three rounds of rough alignments with PEET[7,68]. Major components, such as radial spokes, nexin-dynein regulatory complex and inner junction filament holes, are well resolved in the 96-nm average (Supplementary Fig. 4). To further improve the resolution, particle coordinates for 48-nm repeat units were generated from the coordinates of the aligned particles of 96-nm repeat units. While keeping all coordinates for 96-nm repeat units, extra coordinates for 48-nm repeat units were added unidirectionally from the 96-nm-repeat-units coordinates in a distance of 48 nm along the axonemal axis. PEET alignments were performed on the particles of 48-nm repeat units to generate the final maps. In total, the sub-tomographic averages of flagellar components from four datasets were generated from the following number of particles: 1014 particles from 20 tomograms of Control dataset 1, 728 particles from 18 tomograms of *FAP106* KD dataset, 1936 particles from 52 tomograms of *MC8* KD dataset, and 538 particles from nine tomograms of Control dataset 2. The cryoET sub-tomographic averaged maps have been deposited in the Electron Microscopy Data Bank (EMDB) under the accession codes EMD-28802, EMD-28803, EMD-28804, and EMD-28805, for *FAP106* KD, Control 1, *MC8* KD, and Control 2, respectively. The resolution of each sub-tomographic average was calculated by *calcUnbiasedFSC* in *PEET* based on the 0.143 FSC criterion (Supplementary Fig. 2).

In addition to averaging particles across all tomograms to obtain high-resolution structures, sub-tomographic averages were performed on individual tomograms having more than 50 particles to assess age-dependent structural differences (Supplementary Fig. 9). While of limited resolution, these analyses identified individual flagella with or without a clear ponticulus, which marks mature flagella[39], and confirmed that FAP106-dependent changes are not a consequence of age-dependent changes (Supplementary Fig. 9).

UCSF ChimeraX[69] was used to visualize reconstructed tomograms and sub-tomographic averages. For surface rendering with UCSF ChimeraX, maps were first low pass filtered to either 20 Å or 30 Å. Assignment of MIP identities in Figs. 2 and 4 is described in Supplementary Fig. 3. Briefly, the densities corresponding to conserved MIPs were assigned colors based on similarity to published models[13,15,17,21]. Additional, FAP106-dependent lineage-specific densities were colored to highlight changes between the control and knockdown structures. Coloring was done with color zone in ChimeraX, and densities were assigned with colors according to the closest structure in the published atomic model PDB:7rro. Model prediction of MC8 was performed by AlphaFold Colab[47–49]. AlphaFold predicted model of MC8 was superimposed into our averaged density with the molmap and fitmap functions in UCSF ChimeraX.

### TMT labeling and quantitative proteomics

Detergent-extracted flagellum skeletons were prepared from $2 \times 10^8$–$4 \times 10^8$ log phase cells as described above. After detergent extraction and depolymerization of subpellicular microtubules, the pellet of flagellum skeletons was washed in Extraction buffer without NP40 and solubilized in 100 μl 8 M urea, 100 mM Tris, pH 8 for 15 min at room temperature. Samples were centrifuged at $16,000 \times g$ for 15 min to pellet any insoluble material. The supernatant was diluted with 100 mM Tris, pH 8 to a final concentration of 2 M urea and precipitated with 20% ice-cold trichloroacetic acid (TCA) on ice overnight. The sample was centrifuged at $16,000 \times g$ for 30 min at 4 °C and washed twice with −20 °C acetone. The TCA-precipitated pellets were dried and stored at ≤−20 °C.

**Sample digestion.** Protein pellets were resuspended with digestion buffer (8 M urea, 0.1 M Tris-HCl pH 8.5). Each sample was normalized by absorbance at 280 nm and 25 μg was aliquoted for the digestion. Samples were reduced and alkylated in digestion buffer via sequential 20-min incubations with 5 mM TCEP and 10 mM iodoacetamide at room temperature in the dark. Carboxylate-modified magnetic beads (CMMB) also known as SP3[70] and ethanol (50% final concentration) were added to each sample to induce protein binding to CMMB beads. CMMB beads were washed three times with 80% ethanol and then resuspended with 18 μl of 100 mM TEAB. Protein samples were digested overnight with 1 μl of 0.1 μg/μl LysC (Promega) and 2 μl of 0.4 μg/μl trypsin (Pierce) at 37 °C.

**TMT labeling and CIF fractionation.** Nine μl of 100% acetonitrile were added to each sample to a final concentration of 30% (v/v). Then the TMT labels were resuspended with 100% acetonitrile. In total, 25 μg of each sample was labeled using TMT6plex (Thermo Fisher Scientific) and the four labeled samples of each set (two independent biological replicates each of control and knockdown) were pooled. The pooled samples were fractionated by CMMB-based Isopropanol Gradient Peptide Fractionation (CIF) method[71] into six fractions before mass spectrometry (MS) analysis.

**LC-MS acquisition and analysis.** Fractionated samples were separated on a 75 μm ID × 25 cm C18 column packed with 1.9 μm C18 particles (Dr. Maisch GmbH) using a 140-min gradient of increasing acetonitrile and eluted directly into a Thermo Orbitrap Fusion Lumos mass spectrometer where MS spectra were acquired using SPS-MS3. Protein identification was performed using MaxQuant[72] v 1.6.17.0. against a user assembled database consisting of all protein entries from the TriTrypDB (https://tritrypdb.org/tritrypdb/app)[73] for *T. brucei* strain 927 (version 7.0). Searches were performed using a 20 ppm precursor ion tolerance. TMT6plex was set as a static modification on lysine and peptide N terminal. Carbamidomethylation of cysteine was set as static modification, while oxidation of methionine residues and N-terminal protein acetylation were set as variable modifications. LysC and Trypsin were selected as enzyme specificity with maximum of two missed cleavages allowed. One percent false discovery rate (FDR) was used as a filter at both protein and PSM levels. Statistical analysis was conducted with the MSstatsTMT Bioconductor package, which uses a moderated *t*-test with Empirical Bayes (variance) moderation of the standard errors and then adjusts the *p* values to account for multiple comparisons by the method of Benjamini–Hochberg FDR[74].

### Reporting summary

Further information on research design is available in the Nature Portfolio Reporting Summary linked to this article.

## Data availability

The structure data generated during the current study have been deposited in the Electron Microscopy Data Bank (EMDB) repository, with the accession codes EMD-28802 (*FAP106* KD), EMD-28803 (Control 1), EMD-28804 (*MC8* KD), and EMD-28805 (Control 2). The previously published structure of the DMT from *Bos taurus* shown in Supplementary Fig. 3 is available in the EMDB repository under the accession code EMDB-24664 and the atomic model is available as PDB entry 7rro[13] (https://doi.org/10.2210/pdb7RRO/pdb). The raw proteomics data generated during the current study have been deposited to the Mass Spectrometry Interactive Virtual Environment (MassIVE) repository under accession IDs MSV000090661 (https://doi.org/10.25345/C5M32NF6G) (TMT proteomics) and MSV000090660 (https://doi.org/10.25345/C5QV3C80X) (APEX2 proximity proteomics), and the analysis tables are provided as Supplementary Data 1 and 2, respectively. *T. brucei* genome and protein sequences are publicly available on TriTrypDB[73] (https://tritrypdb.org/tritrypdb/app) and protein localizations are publicly available on TrypTag[44] (http://tryptag.org). Source data are provided with this paper.

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

## Acknowledgements

We thank Daniel Velez-Ramirez for technical assistance with APEX2 proximity proteomics and Simon Imhof for helpful discussions. We also thank Astra Bryant for assistance with high speed microscopy. Funding was provided by NIH grants AI052348 (K.L.H.), GM071940 (Z.H.Z.), and GM089778 (J.W.). K.J. was supported by the Beckman Scholars Program (Beckman Foundation). We acknowledge the use of resources in the Electron Imaging Center for Nanomachines supported by UCLA and grants from NIH (S10RR23057 and S10OD018111) and NSF (DBI-1338135 and DMR-1548924).

## Author contributions

M.M.S., A.S.W., H.W., J.Z., Z.H.Z., and K.L.H. designed research. M.M.S., A.S.W., H.W., J.Z., J.S., N.S., S.V., P.P., and K.J. performed research. M.M.S., A.S.W., H.W., J.Z., J.S., N.S., S.V., P.P., K.J., J.W., Z.H.Z., and K.L.H. analyzed data. M.M.S., A.S.W., H.W., Z.H.Z., and K.L.H. wrote the paper.

## Competing interests

The authors declare no competing interests.
