## [Peer Review File · Nature Communications]

REVIEWER COMMENTS

Reviewer #1 (Remarks to the Author):

Recent cryo-EM studies revealed a number of proteins binding inside the lumen of microtubules in cilia, called MIPs. They form intriguing conformation – some highly coiled-coil and located along microtubule protofilaments, others transversely bridging between tubulins belonging to multiple protofilaments. While high resolution structures of these MIPs were analyzed by single particle cryo-EM (Ma et al. (2019) Cell 179, 909), which identified them in a recent few years, our knowledge about their function is limited. Some MIPs (FAP45 and FAP52) were shown to stabilize protofilaments bundled to each other and make motile cilia robust enough during beating (Owa et al. 2019). We would like to know if the role of MIPs is limited to such a structural one, or there are more active functions. This work, Shimogawa et al., investigated cilia from parasite Trypanosome brucei and characterized one MIP, FAP106, by interdisciplinary approach with RNAi knockdown, cryo-electron tomography (cryo-ET) and APEXs proximity proteomics.

They proved by cryo-ET that FAP106 is located at the inner junction of microtubule doublets. FAP106 knockdown strain shows shortening of cilia and decrease of motility as well as loss of other MIPs, PACRG-B, FAP45 and FAP210, the positions of which were indicated by cryo-ET. Moreover, they found other proteins, which were not known as MIPs, are also affected by FAP106 knockdown. They employed APEX2 and located them to be in proximity to ciliary microtubule protofilaments. Especially one of these proteins, MC8, is identified as a pyramid-shape MIP inside the B-tubule.

The manuscript and the figures are prepared clearly in a convincing way. This reviewer is confident that wide readers, not only cilia researchers, will be interested in this work, and thus recommends publication of this manuscript in Nature Communications after minor revision.

Comments:

Line79-86 “FAP106 is required for parasite motility”

Based on their data, it seems not excluded that FAP106 is critical for flagellar elongation and the length of flagella is essential for normal parasite motility, instead of FAP106 directly functions for motility. Or do they have any evidence to exclude this possibility? Is there any mutant which keeps flagellar length similar to FAP106 KD, but maintains normal motility?

Line108 “FAP45 N-terminus are reduced ... (Fig.2D orange)”

This density change is not clear in Fig.2D, because the orange part is behind the cyan one in Fig.2D Ctrl (in the movie it is clearly presented). Please fix it with alternative view and/or magnification.

Line124-130 Six holes per 96-nm unit

This is a subtle issue which could be influenced by artifact. Although we know most (or all) known MIPs follow 48nm periodicity, it is not guaranteed that PACRG-B also does. This reviewer would recommend the authors whether average with 96nm repeat gives the same conclusion. If the 96nm average is too noisy for surface rendering representation, cross section can be used.

Line157-165 APEX2-based proximity proteomics

The authors identified four proteins (Tb927.10.7120, Tb927.11.4920, Tb927.11.2770, Tb927.3.3200) to be reduced upon knock-down of FAP106, based on tandem mass-tagged proteomics. They further employed APEX2-based proximity proteomics analysis and identified them to be novel MIP candidates and named them MC3, MC5, MC8 and MC15. Since this reviewer is not a proteomics expert, this APEX2 identification process is not clear and afraid that it is similar to many readers. Probably the necessary information is involved in the supplementary data 1&2, but further explanation will help for the readers to follow how these proteins were identified as MIPs. How were they named (do the numbers 3, 5, 8 and 15 have any meaning)?

Reviewer #2 (Remarks to the Author):

The authors used cryo-ET to reveal the functions of FAP106/ENKUR, a microtubule inner protein (MIP) at the doublet microtubule (DMT) within *T. brucei* cilia. The comparative cryoET clearly showed FAP106 is required for assembly of FAP45 and some other MIPs in B-tubule. The authors also applied TMT-based quantitative proteomics to identify FAP106-dependent MIPs, and four previously uncharacterized proteins were identified as MIP candidates. The independent APEX2-based proximity proteomics was also conducted to validate these MIP candidates. Finally, one of candidates, MC8, was found to be a *T. brucei* MIP required for parasite motility.

FAP106 is a homolog of Enkurin (ENKUR) which is a conserved and important protein for sperm motility in many species (Sutton, K.A., et al., *Dev Biol*, 2004; Jungnickel, M.K., et al., *Biol Reprod*, 2018). FAP106 is also known to form a 'tether loop' that is essential for structural stability of the DMT (Khalifa, AAZ., et al., *Elife*, 2020). The interactions and the functional study of FAP106 could promote the mechanism understanding for MIP assembly.

Overall, the results from this manuscript are clearly presented and convincing. However, there are some issues that need to be addressed. I would like to make some comments on the proteomics part of the manuscript, while leaving the cryo-ET aspect to other specialists.

Major points:

1. For the APEX2-based proximity proteomic analysis, spectral count-based label free quantification (LFQ) was conducted based on only two independent biological replicates. I am afraid the LFQ reproducibility issue may lead to the false result. Could the authors provide some quality control analysis, such as Pearson's correlation graphs? It is good practice to perform biological assays in triplicate to assess the variability of the method. I suggest that the authors provide volcano plots to reveal statically changed proteins in FAP45-APEX and FAP52-APEX cells.

2. For the identification of MIP candidates via APEX2-based proximity labeling, the authors set 4 filtering criteria: "i. higher enrichment in MIP (FAP45-APEX and FAP52-APEX) samples vs. non-MIP control sample (DRC1-APEX) compared to known B-tubule MIPs, ii. relative abundance comparable to or greater than known B-tubule MIPs, iii. no homolog in Chlamydomonas or Tetrahymena based on OrthoMCL and/or BLAST search, iv. not previously characterized (annotated as hypothetical or domain of unknown function)." However, more quantitative descriptions should be given to clarify the data filtration process. I suggest the authors add additional columns about the match of these criteria for each identified protein (in Supp. Dataset S2).

Minor points:

1. The citation of Reference #3 seems to not relevant to "eukaryotic pathogens responsible for tremendous human suffering worldwide".

2. Reference #43 and #60 need to be updated.

3. Line 324. The abbreviation DPBS was defined twice.

4. Line 385, "The sample was centrifuged at full speed..." The centrifugal force needs to be specified.

5. Line 391 and 769. Since the pH is critical for the alkylation reaction, the buffer for alkylation should be specified.

6. Line 752, "DPBS + 8% NDS + 2% BSA". The abbreviation NDS should be defined.

Reviewer #3 (Remarks to the Author):

This piece of work aims to analyse the proteins which decorate the interior of the B tubule of the microtubule outer doublets of flagellar axoneme, specifically considering the human parasite *T. brucei*. This is a question of central importance to the structure of the axoneme, which is found across eukaryotic life and vital for cell motility in many systems.

The system uses a combination of RNAi knockdown, cryo electron tomography and mass spectrometry to characterise knockdown of a known microtubule inner protein (MIP) to map an assembly hierarchy and identify a lineage specific MIP necessary for *T. brucei* swimming. However, the study fails to take into account a known feature of the cell biology of *T. brucei* axonemes - that there are B tubule projections incorporated into the assembled axoneme as the flagellum matures (10.1016/j.cub.2006.05.041).

Overall, the data shown is mostly compelling and largely supports the major conclusions. However I have some significant concerns.

As the cryo-ET data capture appears not to have taken into account known flagellum-age dependent structural changes in the B tubule lumen, ie. identifying flagella by age from length and/or associated cytoskeleton remnant morphology, the averaged structures have an unknown bias towards the new or mature B tubule lumen organisation. This is visible in the data: concerningly, the controls in Figure 2 and 4 significantly differ in the B tubule lumen (when they'd be expected to be the same), suggesting this bias is indeed an issue. Furthermore, this means that it is not clear that the changes in cryo ET structure between the controls and RNAi KD cell lines are entirely due to the KD. I also have some concerns with interpretation of protein identity from the cryo ET structure. Overall, the proteomics data appear robust, and support the conclusions from cryo ET despite the limitations in assay design. One key control for accurate knockdown for several cell lines appears to be missing.

Figure design is largely good, showing the magnitude of effect, replicate variability and statistical tests well. Some would benefit from showing the data points from individual replicates, eg. Figure 1A, 1E, 3D. Supplemental figure design is inadequate, leaving me unable to evaluate several pieces of evidence - see specific points below.

There is some interesting discussion of the eukaryote-wide implications for how MIPs and related structures assemble. However, I find the discussion somewhat speculative, focusing on speculative points while it seems to miss some key important discoveries which, while involving parasite-specific proteins, may speak to important and previously un-analysed microtubule lumen decoration dynamics (see below).

Introduction.

Perhaps around Line 41. Some introduction of the post-axoneme assembly of *T. brucei ponticuli* (B tubule luminal projections, 10.1016/j.cub.2006.05.041) should be made. It is likely this work greatly informs the nature of this past discovery, see my later comments. It also should have influenced how cryo-ET data was captured and is interpreted.

Line 74: Be careful with the term "control". You provide evidence that these proteins are necessary for cell swimming, not that they control cell swimming. Failure to move is a failure to be able to move, which may be a catastrophic failure of generating a coordinated flagellar beat rather than mis-control of motility.

Line 75. If the true impact of this work is in therapeutic development then it would be important to introduce the more important human pathogens in this family, *Leishmania* and *Trypanosoma cruzi*. However, in reality, I have serious doubts that structural components of a flagellum - whether or not they are proteins specific to a parasite - are viable therapeutic targets.

Results.

Section starting line 90.

1) How were the flagella selected for Cryo-ET? Is it known whether they are the old or new flagellum, when looking at cells with two flagella? The structure in Figure 2 looks somewhat unlike the mature flagellum, based on axoneme cross-sections viewed by conventional chemical fixed and osmium tetroxide/uranyl acetate/lead acetate-stained EM samples. This old methodology shows very strong electron density in cells with a mature flagellum, spanning protofilament 12 of the A tubule to 4/6 of the B tubule, by your numbering in Fig 2. Some comment on this disparity, whether it comes from bias to new flagella in the cryo-ET samples, potential high stain affinity of the projections in this regions by classical EM methods, etc. is important. Note that the electron density in this region for the control and MC8 knockdown (Figure 4C) significantly differ from the control in Figure 2.

2) I find it unclear how the molecular identity of MIPs was assigned. It is stated that "referencing the position of MIPs conserved in other organisms provides insight." with very little further detail, and the methods section seems to have no information at all. I can see structural similarities to other organisms, however, as one example, how can PACRG-A and B be identified as an alternating repeat? And MIPB8 as distinct from PACRG-A/B beyond pure assumption? Perhaps I am misunderstanding, but I think much more exposition is needed to explain the assignment. Clarity is needed to avoid interpretations being taken as fact.

Section starting line 155. This section does not mention/notice the clear cell cycle-dependent localisation of Tb927.3.3200. I cannot see the data to evaluate Figure S3 (the images are far too small, there is no DNA stain), however the genome-wide localisation project (TrypTag) identified this

as a cell cycle-dependent axoneme component by C terminal tagging (<http://tryptag.org/?query=Tb927.3.3200>) which localises only to the newly forming flagellum. This contrasts ponticuli which are absent in newly forming flagella and assemble later into the mature axoneme. This is important for two reasons:

- 1) It is an example of microtubule lumen remodelling mapped to a specific protein, a protein which is lost from a fully assembled B tubule, and
- 2) It occurs in the inverse to gain of ponticuli, suggesting a potential inhibitory role of Tb927.3.3200 against the formation of ponticuli.

While not Tb927.3.3200 was not mapped to an individual projection, this is an important discovery that needs discussion.

Section starting line 166. Is evidence for knockdown of the expected target confirmed for MC3, 9 and 15? I could not find this important control experiment.

Section starting line 180. It is important to also do a search for proteins with a structural similarity to MC8/MIP B8, to detect proteins with highly divergent sequence yet similar structure but an analogous role. NB. Using an improved AlphaFold prediction (see below, http://wheelerlab.net/alphafold/TbruceiTREU927/view.php?idse=Tb927.11.2770_1-270) and FoldSeek I was unable to find any proteins with a similar predicted structure beyond Euglena, including no hit in Tetrahymena (see line 247).

Figures.

Figure 1. The insets in C and D are far too small to be useful - these should either be removed or much larger.

Figure 2. I'm struggling to find MIP B5 in the figure, is this not shown in the sections C-E?

Figure 4. Note that for A, an improved prediction can be obtained using AlphaFold and a custom multiple sequence alignment (10.1371/journal.pone.0259871).

Figure 5. The trypanosome and movement diagrams are, I'm afraid, quite poor. Especially as the small but statistically significant decrease in flagellum length is not represented, and the movement shown arguably misrepresents the near-complete absence of movement as undirected but fast movement. The summary diagram also does not accurately show the retained projection around microfilaments 11/12 of the A tubule. Please consider thicker line weights and darker colours in the microtubule diagram to make the proteins more visible.

Supplemental figures.

The supplemental figures are largely poorly prepared and hard to interpret/see.

Figure S1. Text is too small to read on axis labels.

Figure S2. This graph lacks axis labels or units, and is clearly a default Excel graph with an absence of effort for clear data presentation.

Figure S3. The microscopy images are too small and too low contrast to be visible. The text, particularly in C-G is too small to read.

Figure S4. I am colour blind and cannot see the merge in B. Fluorescent channels should ideally be shown both separately in greyscale and merged with colour blind-safe colours, eg. green and magenta.

Discussion.

I find the discussion overall interesting, considering conserved aspects of flagellum assembly. However it egregiously lacks any discussion of the new flagellum-specific MIP, which must be disassembled from assembled B tubules. Nor does it link this with the assembly of the ponticulus into the B tubule. Nor does it explain this as a potential study limitation, nor link this with the apparent differences between the controls in Figure 2 vs 4.

REVIEWER RESPONSE

We thank all three reviewers for their careful reading of the manuscript and thoughtful comments. In response, we have carried out new experiments, performed additional data analyses, and revised the manuscript by incorporating these new data and improving clarity of the manuscript. As you can see from the point-by-point responses below, all issues raised by the reviewers have been addressed through these efforts, which have greatly strengthened the work. For your convenience of perusal, we have copied the reviewers' comments in **black** and our point-by-point responses to reviewer comments are in **blue** below.

Reviewer #1 (Remarks to the Author):

Recent cryo-EM studies revealed a number of proteins binding inside the lumen of microtubules in cilia, called MIPs. They form intriguing conformation – some highly coiled-coil and located along microtubule protofilaments, others transversely bridging between tubulins belonging to multiple protofilaments. While high resolution structures of these MIPs were analyzed by single particle cryo-EM (Ma et al. (2019) Cell 179, 909), which identified them in a recent few years, our knowledge about their function is limited. Some MIPs (FAP45 and FAP52) were shown to stabilize protofilaments bundled to each other and make motile cilia robust enough during beating (Owa et al. 2019). We would like to know if the role of MIPs is limited to such a structural one, or there are more active functions. This work, Shimogawa et al., investigated cilia from parasite Trypanosome brucei and characterized one MIP, FAP106, by interdisciplinary approach with RNAi knockdown, cryo-electron tomography (cryo-ET) and APEXs proximity proteomics.

They proved by cryo-ET that FAP106 is located at the inner junction of microtubule doublets. FAP106 knockdown strain shows shortening of cilia and decrease of motility as well as loss of other MIPs, PACRG-B, FAP45 and FAP210, the positions of which were indicated by cryo-ET. Moreover, they found other proteins, which were not known as MIPs, are also affected by FAP106 knockdown. They employed APEX2 and located them to be in proximity to ciliary microtubule protofilaments. Especially one of these proteins, MC8, is identified as a pyramid-shape MIP inside the B-tubule.

The manuscript and the figures are prepared clearly in a convincing way. This reviewer is confident that wide readers, not only cilia researchers, will be interested in this work, and thus recommends publication of this manuscript in Nature Communications after minor revision.

We are pleased that the reviewer recognizes the importance of the work, finds the manuscript and figures clear and compelling, and supports publication of the manuscript. We have made the minor revisions suggested by the reviewer, and the revised manuscript is strengthened as a result. See point-by-point responses below.

Comments:

Line79-86 “FAP106 is required for parasite motility”

Based on their data, it seems not excluded that FAP106 is critical for flagellar elongation and the length of flagella is essential for normal parasite motility, instead of FAP106 directly functions for motility. Or do they have any evidence to exclude this possibility? Is there any mutant which keeps flagellar length similar to FAP106 KD, but maintains normal motility?

To our knowledge, the effect of shortened flagella on motility has not been carefully investigated in *T. brucei*. We agree that our data cannot distinguish between a direct role for FAP106 in motility and an indirect effect due to loss of additional MIPs and/or the reduced length of flagella in the FAP106 KD, and we did not intend to imply that we've shown a direct requirement. In either case, the function of FAP106 is required (directly or indirectly) for normal parasite motility, so we think the statement is valid, but have clarified the text to indicate that the requirement may be direct or indirect. See lines 92-93.

Line108 “FAP45 N-terminus are reduced ... (Fig.2D orange)”

This density change is not clear in Fig.2D, because the orange part is behind the cyan one in Fig.2D Ctrl (in the movie it is clearly presented). Please fix it with alternative view and/or magnification.

The reviewer makes a valid point. Although the cryoET data are consistent with loss of FAP45 (as supported separately by proteomics results), the current resolution does not clearly resolve filamentous densities such as that expected for FAP45, making this a challenging point to make based purely on the cryoET structure. We have therefore removed references to loss of FAP45 densities in this figure (Lines 119-121) and only reference the loss of FAP45 after definitively demonstrating loss of the protein by TMT proteomics (Lines 157-160).

Line124-130 Six holes per 96-nm unit

This is a subtle issue which could be influenced by artifact. Although we know most (or all) known MIPs follow 48nm periodicity, it is not guaranteed that PACRG-B also does. This reviewer would recommend the authors whether average with 96nm repeat gives the same conclusion. If the 96nm average is too noisy for surface rendering representation, cross section can be used.

As requested, we have performed 96nm subtomographic averages, and have included this as a supplemental figure (Supp. Fig. S4), showing that there are indeed six holes per 96-nm unit in the FAP106 KD.

Line157-165 APEX2-based proximity proteomics

The authors identified four proteins (Tb927.10.7120, Tb927.11.4920, Tb927.11.2770, Tb927.3.3200) to be reduced upon knock-down of FAP106, based on tandem mass-tagged proteomics. They further employed APEX2-based proximity proteomics analysis and identified them to be novel MIP candidates and named them MC3, MC5, MC8 and MC15. Since this reviewer is not a proteomics expert, this APEX2 identification process is not clear and afraid that it is similar to many readers. Probably the necessary information is involved in the supplementary data 1&2, but further explanation will help for the readers to follow how these proteins were identified as MIPs. How were they named (do the numbers 3, 5, 8 and 15 have any meaning)?

Thank you for pointing out this lack of clarity regarding details of the APEX proteomics analysis (also noted by Reviewer 2). In response, we have revised the manuscript to provide a clearer explanation in the main text (Lines 182-188), as well as in Supplementary Methods and Supplementary Dataset S2. Briefly, using APEX2 proximity labeling, 15 proteins were identified as putative MIP candidates based on proximity to two known MIPs, FAP45 and FAP52 and additional criteria summarized below (Supp Dataset S2). APEX proximity labeling identifies proteins in proximity to an APEX-tagged protein, "the bait", based on biotinylation and subsequent streptavidin purification and proteomic identification. We used as bait, two MIP proteins known to be inside the B-tubule of the DMT, FAP45 and FAP52, and one axonemal protein known to be outside the DMT, DRC1. We then compared relative abundance for proteins in the MIP vs DRC1 bait samples. The threshold abundance ratio for inclusion as a MIP candidate is based on the average FAP45/DRC1 and FAP52/DRC1 ratios obtained for known MIPs. Additional filters were subsequently applied to identify the most promising MIP candidates (MCs). In summary, proteins are identified as MCs based on: i) average FAP45/DRC1 and FAP52/DRC1 ratio meeting MIP threshold in APEX2 proteomics experiments, ii) similar abundance to known MIPs in APEX2 proteomics experiments, iii) localization to the flagellum from published data, iv) absence of prior functional annotation, and v) lack of homologs outside kinetoplastids. The 15 MCs identified based on these criteria were named MC1-15 in no particular order, and four of these MCs (MC3, 5, 8, and 15) were subsequently independently identified in a separate TMT proteomics experiment as being reduced in FAP106 KD relative to controls. The exact filtering criteria are specified in the Supplementary Methods (Lines 949-980) and in Supplementary Dataset S2 (and detailed below in response to Reviewer 2).

Reviewer #2 (Remarks to the Author):

The authors used cryo-ET to reveal the functions of FAP106/ENKUR, a microtubule inner protein (MIP) at the doublet microtubule (DMT) within *T. brucei* cilia. The comparative cryoET clearly showed FAP106 is required for assembly of FAP45 and some other MIPs in B-tubule. The authors also applied TMT-based quantitative proteomics to identify FAP106-dependent MIPs, and four previously uncharacterized proteins were identified as MIP candidates. The independent APEX2-based proximity proteomics was also conducted to validate these MIP candidates. Finally, one of candidates, MC8, was found to be a *T. brucei* MIP required for parasite motility.

FAP106 is a homolog of Enkurin (ENKUR) which is a conserved and important protein for sperm motility in many species (Sutton, K.A., et al., Dev Biol, 2004; Jungnickel, M.K., et al., Biol Reprod, 2018). FAP106 is also known to form a ‘tether loop’ that is essential for structural stability of the DMT (Khalifa, AAZ., et al., Elife, 2020). The interactions and the functional study of FAP106 could promote the mechanism understanding for MIP assembly.

Overall, the results from this manuscript are clearly presented and convincing. However, there are some issues that need to be addressed. I would like to make some comments on the proteomics part of the manuscript, while leaving the cryo-ET aspect to other specialists.

We are pleased that the reviewer finds the results clear and compelling. We also agree with the reviewer's assessment of areas that could be improved, and have further clarified and strengthened the revised manuscript accordingly. Specific points are addressed below.

Major points:

1. For the APEX2-based proximity proteomic analysis, spectral count-based label free quantification (LFQ) was conducted based on only two independent biological replicates. I am afraid the LFQ reproducibility issue may lead to the false result. Could the authors provide some quality control analysis, such as Pearson's correlation graphs? It is good practice to perform biological assays in triplicate to assess the variability of the method. I suggest that the authors provide volcano plots to reveal statically changed proteins in FAP45-APEX and FAP52-APEX cells.

We recognize that there is inherent variability with LFQ. As requested, we have included Pearson's correlation graphs and coefficients for the two biological replicates in Supplementary Figure S6C, showing good correlation ($r \geq 0.95$) between each set of replicates. Given the inherent variability in proteomics data and small number of replicates, we don't think a statistical analysis/volcano plot of the APEX proteomics data will provide further clarity. However, we would like to emphasize that we performed APEX2-based proximity proteomics with two independent biological replicates for each of two different MIPs (FAP45 and FAP52), for a total of four independent experiments. Thus, we expect our filtering criteria to confidently identify MIP candidates, because only proteins that showed enrichment in proximity to both FAP45 and FAP52 were considered as MCs. We realize that the details of these experiments and the filtering criteria were not clear (as noted in point 2 below and also by Reviewer 1), and have clarified the Supplementary Methods and Supplemental Dataset S2. Furthermore, separate FAP106 KD TMT and cryoET analyses provided independent support that a subset of the MCs identified by APEX proteomics correspond to bona fide MIP structures, thus we opted not to dedicate more effort to APEX analysis.

2. For the identification of MIP candidates via APEX2-based proximity labeling, the authors set 4 filtering criteria: “i. higher enrichment in MIP (FAP45-APEX and FAP52-APEX) samples vs. non-MIP control sample (DRC1-APEX) compared to known B-tubule MIPs, ii. relative abundance comparable to or greater than known B-tubule MIPs, iii. no homolog in Chlamydomonas or Tetrahymena based on OrthoMCL and/or BLAST search, iv. not previously characterized (annotated as hypothetical or domain of unknown function).” However, more quantitative descriptions should be given to clarify the data filtration process. I suggest the authors add additional columns about the match of these criteria for each identified protein (in Supp. Dataset S2).

Thank you for pointing out that more clarity is needed regarding details of the APEX proteomics analysis, a point made by Reviewer 1 also. As noted above in response to Reviewer 1, we have revised the manuscript to provide clearer and more quantitative descriptions in the Supplementary Methods (Lines 949-980) and included additional columns in Supp. Dataset S2, as recommended by this reviewer. Note that we inadvertently missed describing one filtering criterion (flagellum localization) that was used, which we have now rectified thanks to the comments of reviewers. Thus, the revised manuscript now refers to five filtering criteria instead of four.

The filtering criteria are described here for the convenience of the reviewers:

- i. Average FAP45/DRC1 and FAP52/DRC1 ratio meeting MIP threshold in APEX2 proteomics experiments: Average of FAP45/DRC1 and FAP52/DRC1 abundance ratios (Avg MIP/DRC1 ratio) ≥ 1.235 , where 1.24 is the lowest average abundance ratio observed for known B-tubule MIPs, considering PACRGs as B-tubule MIPs based on their exposure to the B-tubule lumen, but excluding FAP20, which for unknown reasons did not show enrichment in MIP samples relative to the non-MIP control.
- ii. Relative abundance comparable to known B-tubule MIPs in APEX2 proteomics experiments: Average abundance in FAP45-APEX samples ≥ 0.00126 and average abundance in FAP52-APEX samples $\geq 9.59e-4$, where 0.0013 is the lowest average abundance observed for known B-tubule MIPs (as defined in i.) in the FAP45-APEX samples and $9.59e-4$ is the lowest average abundance observed for known B-tubule MIPs (as defined in i.) in the FAP52-APEX samples.
- iii. Flagellum localization according to TrypTag (Dean et al., *Trends Parasitol*, 2017; Billington et al., *Nat Microbio*, 2023)
- iv. No homolog in *Chlamydomonas* or *Tetrahymena* based on OrthoMCL and/or reciprocal BLAST search
- v. Not previously characterized (annotated as hypothetical or domain of unknown function)

Minor points:

1. The citation of Reference #3 seems to not relevant to “eukaryotic pathogens responsible for tremendous human suffering worldwide”.

We believe the citation is relevant as *Phytophthora* is a eukaryotic pathogen of crops that poses a substantial agricultural threat. Furthermore, the presence of a flagellated gamete stage in this organism suggests conserved aspects of flagellum biology may be informative for understanding and combating these pathogens.

2. Reference #43 and #60 need to be updated.

We thank the reviewer for catching these inadvertent references to the pre-print versions and have updated the citations to the correct publications. Now References 48 and 67.

3. Line 324. The abbreviation DPBS was defined twice.

Corrected. (Line 358)

4. Line 385, “The sample was centrifuged at full speed...” The centrifugal force needs to be specified.

Corrected to specify 16,000 rcf. (Lines 442, 444)

5. Line 391 and 769. Since the pH is critical for the alkylation reaction, the buffer for alkylation should be specified.

The buffer for alkylation has now been specified as digestion buffer (8M urea, 100 mM Tris-HCl, pH 8.5). (Lines 449 and 928-931)

6. Line 752, “DPBS + 8% NDS + 2% BSA”. The abbreviation NDS should be defined.

Corrected to define normal donkey serum (NDS). (Line 912)

Reviewer #3 (Remarks to the Author):

This piece of work aims to analyse the proteins which decorate the interior of the B tubule of the microtubule outer doublets of flagellar axoneme, specifically considering the human parasite *T. brucei*. This is a question of central importance to the structure of the axoneme, which is found across eukaryotic life and vital for cell

motility in many systems.

The system uses a combination of RNAi knockdown, cryo electron tomography and mass spectrometry to characterise knockdown of a known microtubule inner protein (MIP) to map an assembly hierarchy and identify a lineage specific MIP necessary for *T. brucei* swimming. However, the study fails to take into account a known feature of the cell biology of *T. brucei* axonemes - that there are B tubule projections incorporated into the assembled axoneme as the flagellum matures (10.1016/j.cub.2006.05.041).

Overall, the data shown is mostly compelling and largely supports the major conclusions. However I have some significant concerns. As the cryo-ET data capture appears not to have taken into account known flagellum-age dependent structural changes in the B tubule lumen, ie. identifying flagella by age from length and/or associated cytoskeleton remnant morphology, the averaged structures have an unknown bias towards the new or mature B tubule lumen organisation. This is visible in the data: concerningly, the controls in Figure 2 and 4 significantly differ in the B tubule lumen (when they'd be expected to be the same), suggesting this bias is indeed an issue. Furthermore, this means that it is not clear that the changes in cryo ET structure between the controls and RNAi KD cell lines are entirely due to the KD. I also have some concerns with interpretation of protein identity from the cryo ET structure. Overall, the proteomics data appear robust, and support the conclusions from cryo ET despite the limitations in assay design. One key control for accurate knockdown for several cell lines appears to be missing.

Figure design is largely good, showing the magnitude of effect, replicate variability and statistical tests well. Some would benefit from showing the data points from individual replicates, eg. Figure 1A, 1E, 3D. Supplemental figure design is inadequate, leaving me unable to evaluate several pieces of evidence - see specific points below.

There is some interesting discussion of the eukaryote-wide implications for how MIPs and related structures assemble. However, I find the discussion somewhat speculative, focusing on speculative points while it seems to miss some key important discoveries which, while involving parasite-specific proteins, may speak to important and previously un-analysed microtubule lumen decoration dynamics (see below).

We are pleased that the reviewer recognized the central importance of the work and finds the data mostly compelling and largely supportive of the major conclusions. The reviewer raised a concern about age-dependent changes in the ponticulus structure, which is added into assembled flagella (Vaughan, et. al. *Current Biology* 2006. <https://dx.doi.org/10.1016/j.cub.2006.05.041>), and called into question whether structural changes observed in between controls and RNAi KD can be attributed to the KD versus age-dependent changes. Although we inadvertently omitted a discussion of this known age-dependent structural feature, consideration of age-related changes to the axoneme factored directly into our experimental design as discussed here: First, as samples analyzed for structure determination are isolated flagellum skeletons, with cell bodies/remnants removed, we cannot readily determine mature vs nascent flagella a priori. Therefore, individual flagella were chosen randomly for structural analysis, so as to mitigate imposition of a potential bias. Moreover, whenever comparisons were made, we considered it critical to complete side-by-side, identical analyses in parallel on knockdown versus control samples at equivalent cell densities – and that is what was done. Finally, we considered only substantial structural changes, rather than small perturbations that may not be reliably interpreted at the resolutions obtained.

Importantly, multiple, independent lines of empirical analyses support the view that our experimental design achieved the desired goal of discerning knockdown-dependent changes rather than age-related changes. **First:** when comparing side-by-side samples processed identically and run in parallel (e.g. Fig 2 Ctrl vs FAP106 KD; Fig 4 Ctrl vs MC8 KD), one sees that the average structures for knockdown versus control samples (representing hundreds of particles and multiple flagella for each sample) show very little difference in the ponticulus density, the only validated structural marker for old versus nascent flagella (Vaughan, et. al. *Current Biology* 2006. <https://dx.doi.org/10.1016/j.cub.2006.05.041>). The presence of similar ponticulus densities between side-by-side samples argues against the idea that differences are due to age rather than the KD. **Second:** If one considers that the presence of a more prominent ponticulus in the control sample for Fig. 4

experiments indicates more mature flagella compared to the control sample for Fig. 2 experiments, the data then also indicate that there are no other major structural differences that can be attributed to flagellum age, because the rest of the B-tubule is unchanged. **Third:** To assess this question even more directly, we have now additionally completed sub-tomogram averaging from single axonemes, such that the sample can only be either mature or nascent. As these averages are derived from a limited number of subtomogram particles, they are of low resolution, but sufficient to determine whether the axoneme is mature or nascent (based on presence or absence of a ponticulus). In these new analyses (Supp. Fig. S9), densities corresponding to MIP B5, MIP B8, and PACRG-B/IJ filament holes are present in control samples and missing in the FAP106-KD samples, regardless of presence or absence of the ponticulus. Thus, these defects are clearly due to FAP106-KD and are independent of flagellum age. **Lastly:** We conducted *eight independent quantitative proteomic analyses* of knockdown versus control samples *processed identically in parallel* for four different knockdown targets other than FAP106 (Supp. Fig. S8A-S8D). In each of these eight independent analyses, the *protein targeted by RNAi knockdown was the only affected protein out of >2000 proteins identified* in each sample, except for a ribosomal protein (a likely artifact) for one of the RNAi lines (Supp. Fig. S8B). This result provides strong statistical support for our interpretation that differences observed in knockdown versus control represent knockdown-dependent differences rather than simply being flagellum age-related differences. Therefore, the combined data from several independent analyses firmly support the conclusion that differences observed represent knockdown-dependent differences and are not flagellum age-related differences.

Nonetheless, the reviewer point is well taken and we have revised the text (lines 100-104, 212-216, and 389-425) to better reflect the considerations discussed above.

Introduction.

Perhaps around Line 41. Some introduction of the post-axoneme assembly of *T. brucei* ponticuli (B tubule luminal projections, 10.1016/j.cub.2006.05.041) should be made. It is likely this work greatly informs the nature of this past discovery, see my later comments. It also should have influenced how cryo-ET data was captured and is interpreted.

We thank the reviewer for pointing out this unintentional omission. We agree that some discussion of post-axoneme assembly of the ponticulus is appropriate as discussed above. We also point out above that consideration of the ponticulus did factor into how the data were captured and interpreted. While we anticipate that this detail of B-tubule structure will be of interest to trypanosome researchers, we think it a bit too obscure for a general audience. We have therefore opted not to describe this in detail within the introduction, but instead refer to it within the Methods (Lines 389-425) and Results (Lines 100-104 and 212-216) to clarify the collection and interpretation of the data in response to the reviewer's concerns.

Line 74: Be careful with the term "control". You provide evidence that these proteins are necessary for cell swimming, not that they control cell swimming. Failure to move is a failure to be able to move, which may be a catastrophic failure of generating a coordinated flagellar beat rather than mis-control of motility.

The reviewer makes a valid point. Our intent was to encapsulate any differences that disrupt normal cell swimming without implying a complete lack of flagellum motility (paralyzed flagella). We have changed this sentence to "This work advances understanding of MIP assembly mechanisms and provides insight into parasite-specific MIPs that are important for *T. brucei* motility." (Line 75-77).

Line 75. If the true impact of this work is in therapeutic development then it would be important to introduce the more important human pathogens in this family, *Leishmania* and *Trypanosoma cruzi*. However, in reality, I have serious doubts that structural components of a flagellum - whether or not they are proteins specific to a parasite - are viable therapeutic targets.

The impact of the work is larger than therapeutic development, as it also advances efforts to understand fundamental features of the axoneme, an iconic eukaryotic structure that is, as noted by the reviewer, found across eukaryotic life and vital for cell motility in many organisms. We have modified the text (lines 52-55) to

include *T. cruzi* and *Leishmania* as requested. We recognize that it has been common to discount structural proteins as therapeutic targets, given the tremendous success of drugs targeting replication and membrane proteins, however, there are notable success stories targeting structural proteins: for example, a number of drugs targeting the microtubule cytoskeleton have been used clinically, e.g. Colchicine, Taxol, and Vinblastin (Jordan and Wilson, L. Microtubules as a target for anticancer drugs. *Nat Rev Cancer* 4, 253–265, 2004) and beyond microtubules, a prominent example of targeting structural protein is drugs targeting HIV capsid protein (AIDS inhibitor GS-6207).

Results.

Section starting line 90.

1) How were the flagella selected for Cryo-ET? Is it known whether they are the old or new flagellum, when looking at cells with two flagella? The structure in Figure 2 looks somewhat unlike the mature flagellum, based on axoneme cross-sections viewed by conventional chemical fixed and osmium tetroxide/uranyl acetate/lead acetate-stained EM samples. This old methodology shows very strong electron density in cells with a mature flagellum, spanning protofilament 12 of the A tubule to 4/6 of the B tubule, by your numbering in Fig 2. Some comment on this disparity, whether it comes from bias to new flagella in the cryo-ET samples, potential high stain affinity of the projections in this regions by classical EM methods, etc. is important. Note that the electron density in this region for the control and MC8 knockdown (Figure 4C) significantly differ from the control in Figure 2.

We have revised the Methods to better clarify details regarding purification and random selection of flagella for cryoET (Lines 389-398). Because these are purified, demembrated flagella, without a cell body remnant to allow determination of whether they are old/mature or new/nascent/growing flagella, the data represent an unbiased, random sampling of axonemes. As noted by the reviewer, the ponticulus (spanning from protofilament A12 to protofilaments 4/6) is specific to mature flagella. This structure is readily observed in conventional TEM, which looks only at a single flagellum in each image and employs chemical fixatives and contrast-enhancing agents. By contrast, cryoET done here examines native flagella, with no chemical modification, and employs subtomogram averaging of multiple flagella within a mixed population of old and new flagella, thereby limiting ability to clearly resolve the ponticulus, which is only in some of the flagella averaged. Poor contrast in cryoET may also reflect flexibility of the structure in question. We've adjusted the text (Lines 100-104) to clarify differences to published structures and further consideration of mature vs new flagella is addressed as discussed above in response to the first point raised by reviewer 3.

2) I find it unclear how the molecular identity of MIPs was assigned. It is stated that "referencing the position of MIPs conserved in other organisms provides insight." with very little further detail, and the methods section seems to have no information at all. I can see structural similarities to other organisms, however, as one example, how can PACRG-A and B be identified as an alternating repeat? And MIPB8 as distinct from PACRG-A/B beyond pure assumption? Perhaps I am misunderstanding, but I think much more exposition is needed to explain the assignment. Clarity is needed to avoid interpretations being taken as fact.

We thank the reviewer for this helpful feedback. Conserved MIPs (FAP106, FAP52, FAP45) and a FAP210-like density, as well as conserved IJ filament proteins (FAP20 and PACRG) were assigned and colored based on similarity to published *Chlamydomonas* and bovine structures (Ma et al. *Cell*, 2019; Khalifa et al. *eLife*, 2020; Gui et al. *Cell*, 2021). Recently published work on the *Tetrahymena* DMT clearly demonstrates the structural similarities between MIPs across different species (Kubo et al. *Nat Comm*, 2023; Fig 2A), further supporting our assignments based on comparisons to other published structures. To aid in clarifying how MIP assignments were made, we have added a supplemental figure (Supp Fig S3) to directly compare the *T. brucei* structure to prior Bovine structure, in which atomic resolution allowed identification of proteins corresponding to cryoEM densities directly. Lineage-specific densities not present in the *Chlamydomonas* or bovine structures (MIP A12, MIP B5, MIP B8) were colored in the control *T. brucei* sample to highlight densities that were substantially reduced in the FAP106 KD. We have now described this more clearly in the text (Lines 108-111), in the Methods (Line 428-433), and the Fig. 2 legend (Lines 709-712).

Regarding alternating PACRG-A/B: PACRG is a widely conserved protein demonstrated to form the IJ filament, alternating with FAP20 (Dymek et al. *Mol Biol Cell*, 2019). In the case of PACRG-A and B, the assignment of an alternating repeat together with FAP20 is based on the pattern of extra holes in the FAP106 KD IJ filament cryoET structure, in combination with the TMT proteomic data indicating only PACRG-B, but not PACRG-A or FAP20, is reduced in FAP106 KD. Thus, the extra IJ holes are expected to correspond to missing PACRG-B. We have clarified the text (Lines 143-145) and Fig. 2 legend (Lines 724-729) to indicate this is our interpretation of the data, and note that this interpretation is consistent with recent structural studies in *Tetrahymena*, in which PACRG-A and B homologs were resolved and shown to alternate (Kubo et al. *Nat Comm*, 2023).

We are unclear what the reviewer is asking regarding distinction between PACRG-A/B vs MIP B8. The MIP B8 density is located near protofilament B8, which is clearly distinct from the position of PACRG within the IJ filament between protofilaments B10 and A1.

Section starting line 155. This section does not mention/notice the clear cell cycle-dependent localisation of Tb927.3.3200. I cannot see the data to evaluate Figure S3 (the images are far too small, there is no DNA stain), however the genome-wide localisation project (TrypTag) identified this as a cell cycle-dependent axoneme component by C terminal tagging (<http://tryptag.org/?query=Tb927.3.3200>) which localises only to the newly forming flagellum. This contrasts ponticuli which are absent in newly forming flagella and assemble later into the mature axoneme. This is important for two reasons:

- 1) It is an example of microtubule lumen remodelling mapped to a specific protein, a protein which is lost from a fully assembled B tubule, and
- 2) It occurs in the inverse to gain of ponticuli, suggesting a potential inhibitory role of Tb927.3.3200 against the formation of ponticuli.

While not Tb927.3.3200 was not mapped to an individual projection, this is an important discovery that needs discussion.

We appreciate the reviewer's enthusiasm about this protein and the suggestion to consider nuances in localization pattern. Our original fluorescence localization experiments were designed to ask if the protein in question was flagellar, in order to test a prediction for a MIP or an MC, and to assess knockdown. The results, here and in TrypTag, clearly show flagellar localization, and thus are consistent with a MIP/MC assignment, and corroborate TMT proteomics results. Knockdown is also demonstrated by loss of signal upon addition of tetracycline. Having addressed the question posed for these experiments, further analysis for each of the MCs would be the focus of future work. Although TrypTag states cell cycle-dependent localization for MC15, we felt this was not obvious in either our original fluorescence localization analysis (Supp. Fig S5), or in TrypTag. In both cases, the signal is somewhat weak (11th percentile for TrypTag), and no controls were available to rule out impact of differences in focal plane. Nonetheless, in response to reviewer comments, we conducted further analysis using fluorescence microscopy of MC15-neon green (MC15-NG). The results (provided as **Reviewer Figure 1**, pg 11 of Reviewer Response) indicate a complex distribution of signal as summarized here.

To orient reviewers who may not be familiar with *T. brucei* flagellum arrangement, when dividing, *T. brucei* assembles a new flagellum that initiates posterior to and extends alongside the old flagellum (with cell posterior corresponding to the thick end of the cell and anterior corresponding to the flagellum tip). Flagellum assembly initiates at the basal body, which sits adjacent to the kDNA (mitochondrial DNA) that can be visualized with DNA staining.

Summary of results for MC15-NG (**Reviewer Figure 1**): Firstly, all cells and all flagella have NG signal - indicating the protein is NOT lost from a fully assembled B-tubule as suggested by the reviewer; Secondly, in some cases the NG signal is stronger in the distal portion of the flagellum; Third, NG signal is generally stronger in the nascent vs mature flagellum. Co-staining with PFR as a flagellar marker indicates that this is not simply due to position relative to the focal plane. This result is consistent with the possibility that MC15 is enriched in (though not exclusive to) nascent vs mature flagella, though other explanations are also possible. For example, the result might be an artifact, reflecting influence of the NG tag, such that the tagged protein is reduced or replaced by the endogenous, untagged MC15 provided by the untagged allele in these diploid cells.

Deeper investigation is required to determine whether this pattern of localization truly represents removal of the endogenous protein as the flagellum matures. Such studies, and indeed, further analysis of each individual MC identified here, are of interest, but are beyond the scope of the current work. We have therefore restricted our comments regarding fluorescence localization to the context intended, namely asking if they support flagellum localization and successful knockdown.

The reviewer's suggestion of an inhibitory role of MC15 against the formation of ponticuli would be highly speculative, and is inconsistent with current data, namely: i. cryoET data (Fig 2B) do not show increased ponticulus densities when MC15 is reduced in the FAP106 KD (Fig 3A), and ii. TMT quantitative proteomics do not show any proteins to be increased in flagella from MC15 KD (Supp. Fig. 8D).

Section starting line 166. Is evidence for knockdown of the expected target confirmed for MC3, 9 and 15? I could not find this important control experiment.

For each MC, knockdown was assessed and demonstrated both by microscopy showing loss of the mNeonGreen-tagged protein (Supp. Fig S5) and quantitative proteomics data showing significant reduction of the expected target protein in flagella purified from knockdown vs control parasites, with none of the other >2000 proteins significantly changed (Supp. Fig S8). We have made standalone supplementary figures to show this data more clearly and clarified this in the text (Lines 189-192 and 196-201).

Section starting line 180. It is important to also do a search for proteins with a structural similarity to MC8/MIP B8, to detect proteins with highly divergent sequence yet similar structure but an analogous role. NB. Using an improved AlphaFold prediction (see below, http://wheelerlab.net/alphafold/TbruceiTREU927/view.php?idse=Tb927.11.2770_1-270) and FoldSeek I was unable to find any proteins with a similar predicted structure beyond Euglena, including no hit in Tetrahymena (see line 247).

We thank the reviewer for bringing this resource to our attention, and have included citation of this useful tool (Ref. 40). We did attempt to use the original AlphaFold and FoldSeek to identify proteins with divergent sequences but similar structures. As the reviewer notes, we were unable to identify any proteins with structural similarity to MC8 and have now explicitly mentioned this in the Discussion (Lines 277-280).

Figures.

Figure 1. The insets in C and D are far too small to be useful - these should either be removed or much larger.

Removed as suggested.

Figure 2. I'm struggling to find MIP B5 in the figure, is this not shown in the sections C-E?

We apologize for the confusion. MIP B5 is not visible in the views shown in C-E. We have added another longitudinal section showing MIP B5 as Panel F.

Figure 4. Note that for A, an improved prediction can be obtained using AlphaFold and a custom multiple sequence alignment (10.1371/journal.pone.0259871).

Noted. Upon comparison with the original AlphaFold prediction, we find that the two methods predict nearly identical structures for the region of MC8 that can be predicted with high confidence (**Reviewer Fig 2**). We do not draw any conclusions directly from the predicted model; rather, it merely allowed us to make an educated decision to perform cryoET analysis on MC8 KD to definitively determine which cryoET density corresponds to MC8. We have thus elected to keep the original predicted model in the figure. Nevertheless, we have added citations of the Wheeler AlphaFold reference throughout the manuscript to acknowledge the value of this resource (Ref. 49).

Figure 5. The trypanosome and movement diagrams are, I'm afraid, quite poor. Especially as the small but statistically significant decrease in flagellum length is not represented, and the movement shown arguably misrepresents the near-complete absence of movement as undirected but fast movement. The summary diagram also does not accurately show the retained projection around microfilaments 11/12 of the A tubule. Please consider thicker line weights and darker colours in the microtubule diagram to make the proteins more visible.

Thank you for this helpful feedback. We understand the concerns and have made the requested modifications. We think the revised figure suitably represents the lack of directed movement without implying faster movement, and better represents observed MIP densities.

Supplemental figures.

The supplemental figures are largely poorly prepared and hard to interpret/see.

We have expanded the sizes of text and panels, sometimes into standalone figures.

Figure S1. Text is too small to read on axis labels.

Enlarged.

Figure S2. This graph lacks axis labels or units, and is clearly a default Excel graph with an absence of effort for clear data presentation.

Modified as requested.

Figure S3. The microscopy images are too small and too low contrast to be visible. The text, particularly in C-G is too small to read.

Data have been split into three standalone figures to allow enlargement. Microscopy images are now Fig S5. Flagellum length data are now Fig S7. TMT proteomics data are now Fig S8.

Figure S4. I am colour blind and cannot see the merge in B. Fluorescent channels should ideally be shown both separately in greyscale and merged with colour blind-safe colours, eg. green and magenta.

We apologize for this oversight. We have replaced the images as requested. (Now Fig S6B)

Discussion.

I find the discussion overall interesting, considering conserved aspects of flagellum assembly. However it egregiously lacks any discussion of the new flagellum-specific MIP, which must be disassembled from assembled B tubules. Nor does it link this with the assembly of the ponticulus into the B tubule. Nor does it explain this as a potential study limitation, nor link this with the apparent differences between the controls in Figure 2 vs 4.

We are pleased that the reviewer found the discussion interesting. As detailed in the response to reviewer above, omission of reference to the ponticulus was inadvertent and we are grateful that the reviewer caught this important omission. We have now incorporated discussion of the ponticulus and associated limitations, including comments about the apparent differences between the controls (lines 100-104, 212-216, and 389-425). Further comment regarding relevance of expanded discussion of a "new flagellum-specific MIP" is discussed above.

REVIEWER FIGURES

Reviewer Figure 1. Fluorescence microscopy of MC15 (Tb927.3.3200). Anti-PFR (magenta) and Hoescht (blue) are provided as markers of flagella and DNA, respectively. MC15-NG (green) is often stronger in the short, nascent flagellum of a dividing cell than in the old, mature flagellum, however MC15-NG is visible in all flagella and other patterns (such as distal enrichment) are also observed.

Reviewer Figure 2. Comparison of MC8 AlphaFold predictions. **a.** Overlay of our original AlphaFold model (purple) and the improved Wheeler AlphaFold model (brown). **b.** The two models in (a) are shown side by side. The boxed region corresponds to the structure that we attempted to fit into the MIP B8 density in our tomogram (Fig. 4). Although the Wheeler model improves the overall pIDDT score, particularly in the region indicated by the black arrowhead, the regions of folded helices with high pIDDT score are almost identical.

REVIEWERS' COMMENTS

Reviewer #2 (Remarks to the Author):

The authors have addressed my comments. But, this reviewer suggests that the paper should be more carefully evaluated from other aspects as the proximity labeling proteomics part has limited novelty, rather than applying it for exploring proteins.

Reviewer #3 (Remarks to the Author):

I would like to thank the authors for taking care to address my comments in such detail. It is clear that they have taken extensive efforts to address all of my comments - from minor changes in the text and figure presentation to new analyses like Figure S9. I can see all of my major concerns have been addressed well.

Regarding my major comment on the ponticulus: While it would be nice to have some cell cycle dependency information from the cryoEM data, I fully acknowledge the technical difficulty (or possibly impossibility!) there. The single flagella sub-tomogram averages are very informative and address my comments well.

REVIEWER RESPONSE

We thank the reviewers for their careful consideration of the revised manuscript and are pleased they find all their previous concerns adequately addressed. We have copied the reviewers' comments in **black** and our point-by-point responses to reviewer comments are in **blue** below.

Reviewer #2 (Remarks to the Author):

The authors have addressed my comments. But, this reviewer suggests that the paper should be more carefully evaluated from other aspects as the proximity labeling proteomics part has limited novelty, rather than applying it for exploring proteins.

As evidenced in the original response to reviewers, aspects of the work in addition to proximity proteomics have been evaluated rigorously as requested by this reviewer.

Reviewer #3 (Remarks to the Author):

I would like to thank the authors for taking care to address my comments in such detail. It is clear that they have taken extensive efforts to address all of my comments - from minor changes in the text and figure presentation to new analyses like Figure S9. I can see all of my major concerns have been addressed well.

Regarding my major comment on the ponticulus: While it would be nice to have some cell cycle dependency information from the cryoEM data, I fully acknowledge the technical difficulty (or possibly impossibility!) there. The single flagella sub-tomogram averages are very informative and address my comments well.

We thank the reviewer for their careful review, which has led to an improved manuscript.